# Multiphase composition changes and reactive oxygen species formation during limonene oxidation in the new Cambridge Atmospheric Simulation Chamber (CASC)

5 Peter J. Gallimore[1], Brendan M. Mahon[1], Francis P. H. Wragg[1], Stephen J. Fuller[1], Chiara Giorio[1,2], Ivan Kourtchev[1,3] and Markus Kalberer[1]

[1] Department of Chemistry, University of Cambridge, Lensfield Road, Cambridge, CB2 1EW, UK.
[2] now at Aix Marseille Université, CNRS, LCE UMR 7376, 13331 Marseille, France.
10 [3] now at Department of Chemistry, University College Cork, College Road, Cork, Ireland.

*Correspondence to*: Markus Kalberer (mk594@cam.ac.uk)

**Abstract**

The chemical composition of organic aerosols influences their impacts on human health and the climate system. Aerosol formation from gas-to-particle conversion and in-particle 15 reaction was studied for the oxidation of limonene in a new facility, the Cambridge Atmospheric Simulation Chamber (CASC). Health-relevant oxidising organic species produced during SOA formation were quantified in real-time using an Online Particle-bound Reactive Oxygen Species Instrument (OPROSI). Two categories of reactive oxygen species (ROS) were identified based on time series analysis: a short-lived component produced 20 during precursor ozonolysis with a lifetime on the order of minutes, and a stable component which was long-lived on the experiment timescale (~ 4 hours). Individual organic species were monitored continuously over this time using Extractive Electrospray Ionisation (EESI) Mass Spectrometry (MS) for the particle phase and Proton Transfer Reaction (PTR) MS for the gas phase. Many first generation oxidation products are unsaturated, and we observed 25 multiphase aging via further ozonolysis reactions. Volatile products such as $C_9H_{14}O$ (limonaketone) and $C_{10}H_{16}O_2$ (limonaldehyde) were observed in the gas phase early in the experiment, before reacting again with ozone. Loss of $C_{10}H_{16}O_4$ (7-hydroxy limononic acid) from the particle phase was surprisingly slow. A combination of reduced C=C reactivity and viscous particle formation (relative to other SOA systems) may explain this, and both

scenarios were tested in the Pretty Good Aerosol Model (PG-AM). A range of characterisation measurements were also carried out to benchmark the chamber against existing facilities. This work demonstrates the utility of the CASC chamber, particularly for understanding the reactivity and health-relevant properties of organic aerosols using novel, highly time-resolved techniques.

## 1 Introduction

Organic aerosols make an important but poorly understood contribution to the climate system (Boucher et al., 2013). Airborne particles are also closely linked to the negative health effects of air pollution (Pope et al., 2009). Their atmospheric properties, including their interaction with trace gases and ability to act as cloud condensation nuclei, are closely linked to their chemical composition (Abbatt et al., 2012; Hallquist et al., 2009). Detailed chemical speciation is an important step towards understanding the formation and properties of aerosols. In particular, specific compound classes may dominate in certain processes. For example, water-soluble carbonyls may be responsible for a large fraction of aqueous secondary organic aerosol (SOA) formation (Ervens et al., 2011; McNeill et al., 2012). Similarly, species including hydrogen peroxide and oxygen-centred radicals and ions can cause biological stress and damage (Anglada et al., 2015; Apel and Hirt, 2004). Related organic compounds including peroxides have been shown to be major SOA components (Camredon et al., 2007; Docherty et al., 2005). Together, these reactive oxygen species (ROS) are thought to be associated with the observed negative health effects of airborne particles (Verma et al., 2009).

SOA formation is an inherently multiphase process involving both gas-to-particle conversion and heterogeneous and in-particle chemistry (Kroll and Seinfeld, 2008). Atmospheric chambers constitute an invaluable tool for studying these processes under controlled conditions and relevant timescales. A variety of environmental chambers are in use globally, e.g. (Cocker et al., 2001; Hildebrandt et al., 2009; Klotz et al., 1998; Paulsen et al., 2005; Rohrer et al., 2004; Wang et al., 2014) to understand different aspects of atmospheric chemistry, air pollution and chemistry-climate interactions. Previous chamber studies have

led to the discovery of important SOA formation and aging processes (Ehn et al., 2014; Kalberer et al., 2004; Odum et al., 1997; Shiraiwa et al., 2013).

The largest global source of SOA is from the oxidation of biogenic volatile organic compounds (BVOCs) (Hallquist et al., 2009). Limonene is one of the most abundant BVOCs in the troposphere, with an estimated biogenic emission rate of 11 Tg yr$^{-1}$ (Guenther et al., 2012). Its widespread use in industrial processes and household cleaning and fragrance products also results in elevated indoor concentrations with contingent impacts on indoor air quality (Wainman et al., 2000; Waring, 2016; Weschler and Shields, 1999).

Limonene contains two reactive C=C double bonds which results in multiple generations of oxidation products (Bateman et al., 2009; Kundu et al., 2012; Walser et al., 2008) containing a range of functional groups including carboxylic acids, carbonyls, peroxides and alcohols. Previous studies have mainly focused on the reaction of limonene with ozone (Kundu et al., 2012; Zhang et al., 2006), with relatively few OH-aging experiments reported, particularly with respect to chemical characterisation (Zhao et al., 2015). Ozone is a major sink for limonene under a range of atmospheric conditions (Atkinson and Arey, 2003) and will dominate in indoor scenarios which may be most relevant for the health effects of limonene SOA (Waring, 2016). The endo C=C of limonene is more susceptible to ozonolysis by a factor of 10-50 (Zhang et al., 2006) and some of the first-generation ring opening products are condensable (Figure 1). Subsequent oxidation of the remaining double bond may therefore occur in either the gas or condensed phases depending on the properties of the initial products and the aerosol loading.

The ability of limonene to form multifunctional products via successive oxidation steps results in high aerosol yields relative to other terpenes (Hoffmann et al., 1997; Zhang et al., 2006). Aside from ozonolysis, other condensed-phase reactions further modify the composition of limonene SOA. Kundu et al., (2012) report the reactive uptake of carbonyls to form oligomeric products, while the formation of light-absorbing "brown carbon" via uptake and reaction of ammonia and amines appears to be particularly efficient for limonene SOA compared to other precursors (Bones et al., 2010; Updyke et al., 2012).

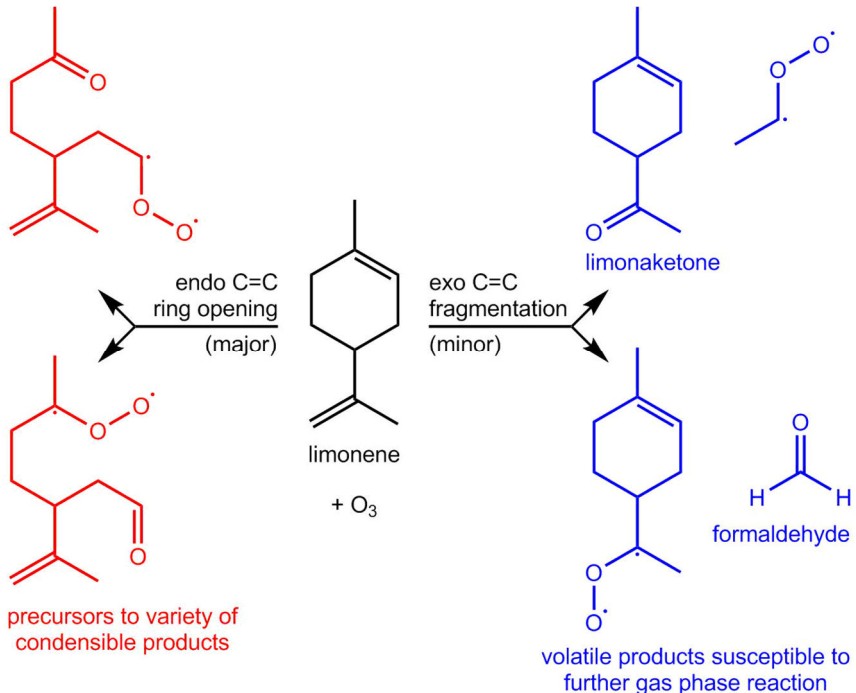

**Figure 1: Initial products of limonene ozonolysis following reaction with the cyclic endo C=C (red channel) and terminal exo C=C (blue channel). The Criegee intermediates produced in these reactions can proceed to form a variety of multifunctional products observed both in the gas and particle phases.**

The multiphase ozone-initiated oxidation of limonene to form SOA was studied in a new, state-of-the-art facility: The Cambridge Atmospheric Simulation Chamber (CASC). CASC is a 5.4 m³ Teflon chamber which is coupled to a range of unique online chemical characterisation instruments. An Extractive Electrospray Ionisation Mass Spectrometer (EESI-MS) provides real-time measurements of particle-phase molecular composition (Gallimore and Kalberer, 2013). EESI retains the key advantage of "soft" electrospray ionisation MS techniques, namely that quasi-molecular ions are produced from aerosol analytes with minimal fragmentation. Individual molecular species can be identified and relative intensity changes monitored over time as a measure of concentration changes with particles (Gallimore et al., 2017). Gas-phase VOC components are monitored using Proton Transfer Reaction Time of Flight (PTR-ToF) MS. Together these complementary techniques produce a detailed, highly time-resolved picture of the evolving organic components in the

chamber on a molecular level. In parallel, an Online Particle-bound Reactive Oxygen Species Instrument (OPROSI) (Wragg et al., 2016) allows the health-relevant oxidising capacity of organic species to be quantified with high time resolution.

In our limonene ozonolysis experiments, we observe several reaction pathways using these instruments which contribute to SOA formation. These include: further oxidation of volatile unsaturated products in the gas phase, heterogeneous reaction of ozone with condensed double bonds, and reactive uptake of carbonyls to form accretion products. Most of the reactive chemistry is complete once the limonene has been consumed, but heterogeneous
reaction and decomposition of ROS appear to continue on longer timescales. We develop model test cases and find that such apparently "slow" rates of change may be explained by a combination of inhibited diffusion within viscous particles and reduced reactivity compared to other SOA systems. Compared to the widely used SOA surrogate, oleic acid aerosol, limonene SOA exhibits a longer ROS lifetime and higher overall ROS yield which we
rationalise in terms of their respective chemical characteristics.

## 2 Methodology

### 2.1 Chamber construction and operation

A schematic of the Cambridge Atmospheric Simulation Chamber (CASC) is given in Figure 2. The design is based on a 5.4 m$^3$ (1.5 × 1.8 × 2.0 m) collapsible bag made from 125 μm
DuPont Teflon fluorocarbon film (FEP type 500A, Foiltec GmbH, Germany). The panels are heat sealed and bonded with Teflon tape (Polyflon Technology Ltd, UK) at the edges and corners of the chamber. The bag is suspended in an aluminium frame (Rexroth, Bosch, Germany) and entirely enclosed by aluminium sheeting and Perspex panels covered with aluminium tape to reflect light. Stainless steel ports containing ¼ " and ½ " stainless steel
bulkheads (Swagelok, UK) are attached to each end of the chamber to enable introduction and sampling of air from the chamber. An initial application of the chamber is described in Kourtchev et al. (2016).

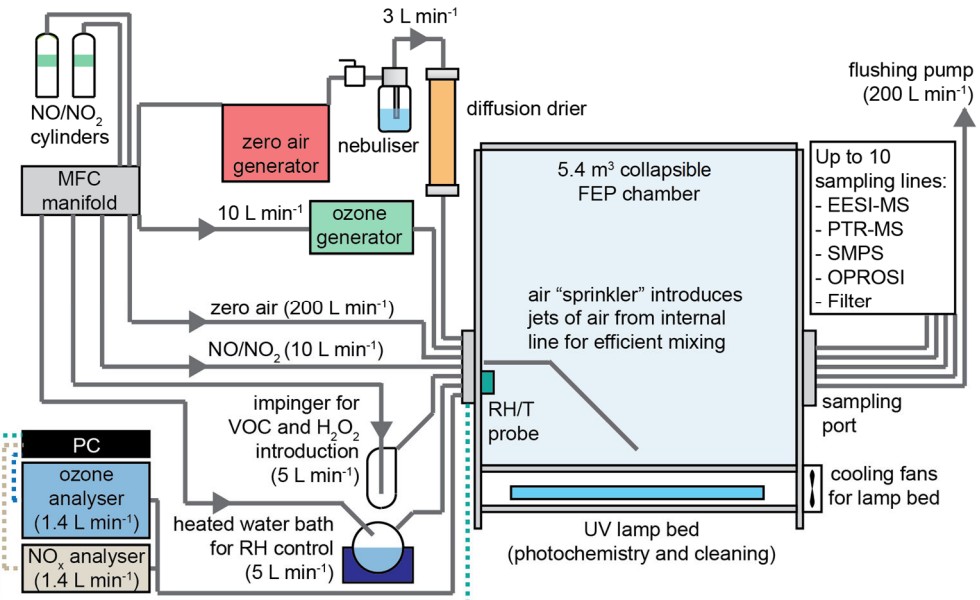

**Figure 2: Schematic of the Cambridge Atmospheric Smog Chamber (CASC). The facility consists of a 5.4 m³ collapsible FEP Teflon chamber with stainless steel ports at each end for introduction of gases and sampling of chamber air by a suite of instrumentation. Gas sampling lines are solid grey, data connections are dashed.**

The temperature of the room which houses the chamber is controlled using a 7.1 kW air conditioning unit (Daikin, UK) and 10 fans are situated underneath one end of the chamber to circulate air from the room through the chamber enclosure. Light sources are situated under the bag and consist of twenty 160 W UV tanning lamps (Philips Body Tone, > 300 nm) for use during photochemical aging experiments and four 75 W "hard" UV lamps (Philips TUV75, 242 nm) for cleaning the chamber. The temperature in the chamber is typically $20 \pm 1$ °C for "dark" experiments and $24 \pm 2$ °C for photochemical aging experiments. Temperature is monitored along with relative humidity (RH) using a probe (Sensirion SHT75, UK) close to the inlet port of the chamber.

The chamber is filled with air from a zero air generator (KA-MT2, Parker Hannifin, UK) which uses a molecular sieve, an activated charcoal bed and filters to remove water, VOCs and particulates respectively. NO and $NO_2$ are supplied from cylinders (each 100 ppm, C

grade, BOC, UK). Flow into the chamber is controlled through a series of mass flow controllers (MFCs) (MKS, UK). Water vapour is introduced by bubbling an air stream through a heated 0.5 L round-bottomed flask containing water (HPLC, Rathbones, UK) which is replaced at least each week. We monitored changes in particle and VOC concentrations during water introduction and found particle formation to be negligible, but observed up to ~ 10 ppb acetone and lesser quantities (< 2 ppb) of $C_1$-$C_2$ aldehydes and acids. Ozone is generated by flowing air through either an enclosed mercury UV lamp (Appleton Woods, UK) or a commercial ozone generator (LABOZONE09, ESCO International Ltd, UK). VOCs and aqueous $H_2O_2$ (if used) are introduced into separate glass impingers and evaporated using an air stream and heating from a heat gun (PHG 2, Bosch, Germany). Seed particles can be optionally introduced from an atomiser (Model 3076, TSI, UK) and are dried using a silica diffusion drier and neutralised with a Kr-85 source (Model 3077, TSI, UK) prior to introduction into the chamber.

A 200 L $min^{-1}$ diaphragm pump (ET200, Charles Austen, UK) is used in combination with 200 L $min^{-1}$ clean air introduction to flush the chamber. Flushing is carried out for at least 24 hours prior to the start of an experiment and may be accompanied by use of the "hard" UV lamps, ozone (~ 10 ppm) and water vapour to remove residual species from the chamber walls. The chamber is typically operated in a batch mode, where reactants are introduced into the chamber at the start of the experiment and allowed to evolve over a period of several hours. Typically a maximum of 1.5 $m^3$ air was removed during a batch experiment, and guide rails on the chamber frame allow the bag to inflate and deflate at ambient pressure. Up to ~ 60 % of the chamber volume could be sampled using this system if required. In principle the chamber could also be operated in a flow-through mode, where continuous introduction of reactants produce steady state conditions in the chamber according to a characteristic mixing time.

**2.2 Chamber instrumentation**

A series of instruments used to monitor physical and chemical parameters of the chamber are listed in Table 1. In addition to a suite of commercially available instrumentation, we also

monitor the chemical evolution of the gas-phase and aerosols formed in the chamber using unique instruments developed in-house.

| Instrument | Measures | Range | Uncertainty | Time resolution |
|---|---|---|---|---|
| EESI-MS (Gallimore and Kalberer, 2013) | Particle-phase chemical composition | 0.2-600 $\mu g/m^3$ | | 4-6 minutes |
| OPROSI (Wragg et al., 2016) | Particle-bound reactive oxygen species (ROS) | 0-2000 nmol [$H_2O_2$] equiv $m^{-3}$ | 2-4 nmol [$H_2O_2$] equiv $m^{-3}$ | 4 minutes |
| Ionicon PTR-ToF 8000 MS | Gas-phase VOCs | | | As low as 100 ms, typically 1 minute |
| TSI 3086 SMPS | Particle size distribution | 14-700 nm | | < 2.5 minutes |
| Thermo 49C ozone analyser | [$O_3$] | 0-200 ppm | $\pm$ 1 ppb up to 1 ppm | 1 minute |
| Teledyne 200E NOx analyser | [NO], [NO2], [NOx] | 0-1000 ppb | $\pm$ 1 ppb | 1 minute |
| Sensirion SHT75 | RH, T | 0-100% (RH), -40 – 120°C (T) | $\pm$ 1.8% (RH), $\pm$ 0.3°C (T) | 1 second |

**Table 1: Overview of CASC instrumentation. EESI-MS and OPROSI are unique instruments**
5 **developed in-house.**

### 2.2.1 Extractive Electrospray Ionisation Mass Spectrometry

Extractive Electrospray Ionisation Mass Spectrometery (EESI-MS) is an online particle analysis technique and the design and optimisation of our EESI source is described in
10 Gallimore and Kalberer (2013). It contains a commercial electrospray probe (Thermo Scientific HESI-II) with a custom-built aerosol injector and manifold. The primary electrospray was operated at a voltage of –3.0 kV and a $N_2$ sheath flow rate of 0.8 L min$^{-1}$. A water−methanol 1:1 mixture (Optima LC-MS grade solvents; Fisher Scientific) containing 0.05% formic acid (90%, Breckland Scientific) was infused into the ESI probe at 10 μL min$^{-}$

[1]. Chamber air was delivered into the source at 1 L min[-1]. Collision and extraction of the SOA particles by primary electrospray droplets occurs in an off-axis configuration with respect to the MS inlet, to minimise source contamination and memory effects through particle deposition. The EESI source was coupled to an ion-trap mass spectrometer (Thermo Scientific LTQ Velos). Spectra were acquired in the negative ionisation mode over the range *m/z* 50-500, with a mass resolution ~ 2000 (full width at half maximum, FWHM) at *m/z* 400.

Gallimore and Kalberer (2013) demonstrated that the relative EESI-MS ion intensity correlated with the mass concentration of tartaric acid particles delivered into the source, suggesting that the entire particle bulk is extracted for analysis. More recently, Gallimore et al., (2017) showed that the kinetics of particle-phase reactions could be monitored; loss rates derived from EESI-MS measurements compared well with other studies, and spectra were compared to Liquid Chromatography (LC) MS to confirm that the EESI-MS assignments were present in the aerosol rather than formed as artefacts in the ion source.

### 2.2.2 Proton Transfer Reaction Mass Spectrometry

The gas phase VOC composition of the chamber is monitored using Proton Transfer Reaction MS (Blake et al., 2009). The PTR-MS (PTR-ToF 8000, Ionicon, Innsbruck, Austria) measures VOCs with a proton affinity higher than water in the *m/z* range 10-500, with a typical mass resolution of 5000 (FWHM) at the mass of protonated acetone, and a typical time resolution of 1 s. Typical detection limits are in the order of 1-2 ppb at 1 s time resolution and ~30 ppt at 1 min time resolution (Blake et al., 2009; de Gouw and Warneke, 2007). For these experiments, source settings were: drift tube voltage of 600 V, drift tube pressure at ~ 2.20 mbar, drift tube temperature at 60ºC, resulting in an E/N of ca. 135 Td (1 Td = $10^{-17}$ V cm$^2$). $k = 2.54 \times 10^{-9}$ cm$^3$ molecule$^{-1}$ s$^{-1}$ was used for limonene quantification (Zhao and Zhang, 2004) and a default rate constant ($k$) of $2 \times 10^{-9}$ cm$^3$ molecule$^{-1}$ s$^{-1}$ was used for the other ions.

### 2.2.3 Online Particle-bound Reactive Oxygen Species Instrument

Reactive Oxygen Species (ROS) can be associated with the negative health impacts of aerosols (den Hartigh et al., 2010; Steenhof et al., 2011). A new Online Particle-bound Reactive Oxygen Species Instrument (OPROSI), described in Wragg et al. (2016) is used to continuously monitor this health relevant property of aerosols from the chamber. The continuous sample inflow (5 L min$^{-1}$) passes through a PM$_{2.5}$ cyclone (URG-2000-30E-5-2.5-S) and charcoal denuder prior to entering into a particle-into-liquid sampler (PILS). Particles are collected into a 1 mL min$^{-1}$ spray containing horseradish peroxidase (HRP) (TypeVI, 1 unit mL$^{-1}$ in 10% phosphate buffer solution (PBS), Sigma Aldrich) which reacts with ROS present in the particles. This is combined with a 1mL min$^{-1}$ aqueous 2'7'-dichlorofluorescein (DCFH) solution (10μM, 10% PBS, Sigma Aldrich), which is oxidised to a fluorescent product (DCF) by the ROS-HRP solution. After a 10 minute reaction time at 40 °C the concentration of DCF is quantified via fluorescence spectroscopy. The fluorescence response is calibrated with H$_2$O$_2$ and quantitative ROS concentrations are reported as "[H$_2$O$_2$] equivalents". The assay also responds to organic peroxides. It is likely sensitive to HO$_x$ radicals and ions such as superoxide but we are unable to obtain suitable standards to test this directly. OPROSI has a time resolution of 4 minutes (e-folding time during online particle collection tests) and is thus able to capture most time-dependant processes observed during SOA formation and evolution. This instrument is especially sensitive to short-lived ROS components, which react within seconds-minutes after sampling (Wragg et al., 2016).

EESI-MS, PTR-MS and OPROSI are all placed in a laboratory just next the room that houses the chamber. Stainless steel tubing (ca. 3 m length) connects the chamber with EESI-MS and OPROSI. The PTR-MS is connected via a 1mm inner diameter PTFE tube kept at room temperature.

### 2.3 Chamber photochemical characterisation

The tanning lamps used during photochemical aging experiments emit primarily in the range 300-400 nm (Figure S1(a)). Emissions below 300 nm, which in the atmosphere are attenuated before reaching the troposphere, are absent. This measured spectrum also overlaps with the absorption cross sections of NO$_2$ (λ < 400 nm) and to a small extent with O$_3$ (λ < 310 nm).

Photolysis of these species drives photochemistry in the troposphere. By contrast, "black lamps", which are commonly employed in chamber studies, emit over a narrower range, 350-400 nm, where ozone photolysis will not occur. Transmission of light through the FEP film used for the bag was tested over the range 200-800 nm (Figure S1(b)). Transmission of light

> 300 nm, used in aging experiments, was higher than 80 %. Transmission of "hard" UV from the cleaning lamps is also acceptable at > 60%.

The photolysis characteristics of the chamber were assessed by quantifying the photolysis frequency of $NO_2$, $J_{NO2}$. Following four $NO_2$ irradiation experiments in which the steady state

concentrations of NO, $NO_2$ and $O_3$ were measured, $J_{NO2} = 0.49 \pm 0.09$ min$^{-1}$ was calculated (Table S1). This is within the range of values determined for other chambers and is comparable to ambient values in Pasadena, California (0.5 min$^{-1}$) (Cocker et al., 2001) and the outdoor EUPHORE chamber (0.44-0.56 min$^{-1}$) (Martın-Reviejo and Wirtz, 2005).

## 2.4 Mixing and wall losses

The chamber air volume is mixed using an "air sprinkler" system. High pressure air is introduced from a PTFE tube (4 mm inner diameter) which extends from the introduction port across the entire the length of the chamber. Periodic holes along the tube allow "jets" of air to escape and mix the chamber volume. This approach avoids the use of mixing fans which may produce unwanted vapours or particles during operation. Mixing with and without

use of the air sprinkler was assessed by evaporating α-pinene into the chamber and monitoring its concentration from the opposite port with the PTR-MS (Figure S2). With 3 × 10 s bursts from the air sprinkler over the course of a minute, the observed α-pinene concentration sampled at the far end of the chamber was seen to stabilise rapidly and reaches 90 % of its steady state value within 4 minutes of mixing. Without active mixing, the α-

pinene concentration took around 30-40 minutes to reach a stable value. This efficient mixing procedure (adding ca. additional 100 L clean air into the chamber) is usually applied during the introduction of oxidants (for a "dark" experiment) or after the addition of all components, before initiating photochemistry (for photochemical aging experiments).

The loss of ozone to the walls of the chamber was tested. Ozone was lost from the clean chamber at an average rate of $5.9 \times 10^{-5}$ min$^{-1}$ over 9 hours. This compares to a loss rate of $1.31 \times 10^{-4}$ min$^{-1}$ for a similar facility, described by Wang et al. (2014).

Particle deposition to the chamber walls was also determined, assuming deposition to be a first order process (Cocker et al., 2001). Ten experiments involving the introduction of ammonium sulfate particles to the clean chamber were performed. Characteristic first-order coefficients for the rate of change of particle number and mass were found to be $\beta_N = 0.201 \pm 0.025$ h$^{-1}$ and $\beta_M = 0.166 \pm 0.020$ h$^{-1}$ respectively, as detailed in the supporting information.

This corresponds to aerosol lifetimes of 5-6 hours, comparable to other chambers as illustrated in Table S2. The potential impact of changes in chamber volume during sampling (maximum ~30 %) on wall loss rates was not considered.

**2.5 Aerosol numerical modelling**

Illustrative model simulations were performed using the Pretty Good Aerosol Model (PG-AM). PG-AM is described in detail in Griffiths et al., (2009) and Gallimore et al., (2017). The model treats the following processes in a kinetic framework: chemical reaction in both the gas and particle phases, gas-particle exchange via uptake and evaporation, and diffusion within the particle. Fluxes between the gas and particle phases depend on the aerosol surface

area as well as each species' accommodation coefficient ($\alpha$, dimensionless) and partitioning coefficient ($K$, M atm$^{-1}$). Diffusion is parameterised according to Griffiths et al., (2009); the particle is treated as a series of nested shells, with the rate of transport of each species between shells determined by its diffusion coefficient ($D$, cm$^2$ s$^{-1}$). The differential equations governing reaction and diffusion are integrated forwards in time using Mathematica (v11,

Wolfram).

In this study, reaction of unsaturated $C_{10}H_{16}O_4$ with ozone was simulated for a single particle of characteristic radius $r_{eff} = 3V_t/S_t = 84$ nm based on the measured total particle volume ($V_t$) and surface area ($S_t$). $C_{10}H_{16}O_4$ was assumed to be 7-hydroxy limononic acid for vapour

pressure calculations and formation of a corresponding carbonyl oxidation product, $C_9H_{14}O_5$,

was represented in the model mechanism (Figure S3). The vapour pressures of $C_{10}H_{16}O_4$ and $C_9H_{14}O_5$ were estimated at $\sim 2.8 \times 10^{-5}$ Pa and $\sim 2.2 \times 10^{-6}$ Pa using the online model EVAPORATION (Compernolle et al., 2011) which performed well in our tests for a species with known vapour pressure, pinic acid. The resulting partitioning coefficients calculated from these vapour pressures, $K \sim 2 \times 10^{10}$ M atm$^{-1}$ and $\sim 2 \times 10^{11}$ M atm$^{-1}$ means that both species are almost entirely condensed (> 99 %) under the experimental conditions.

The accommodation coefficient for organic species was fixed at $\alpha_{org} = 0.1$. Ozone partitioning ($K_{O3} = 0.1$ M atm$^{-1}$) (Morris et al., 2002) and accommodation ($\alpha_{O3} = 10^{-3}$) (Gallimore et al., 2017) were fixed based on the literature for oleic acid particles. The particle phase bimolecular rate constant for ozonolysis ($k_p^{II}$) and the ozone and organic diffusion coefficients ($D_{O3}$ and $D_{org}$) were varied as described in the results. A fixed gas-phase rate constant for exo double bond ozonolysis, $k_g^{II} = 7 \times 10^{-18}$ cm$^3$ molecule$^{-1}$ s$^{-1}$ (Zhang et al., 2006) was also included. However, gas-phase loss was not competitive with particle phase oxidation due to this relatively small rate constant and the low vapour pressure of $C_{10}H_{16}O_4$. The carbonyl product $C_9H_{14}O_5$ was formed with a yield of 0.4, based on the branching of exo-C=C ozonolysis products for β-pinene (Jenkin, 2004). We did not attempt to account for the fate of the other reaction branch featuring a $C_9$ Criegee intermediate due to the wide range of possible Criegee intermediate products in the condensed phase including peroxides, carbonyls and secondary ozonides (Lee et al., 2012; Maksymiuk et al., 2009).

### 3. Limonene SOA formation and characterisation

Insight into the chemical and health-relevant properties of limonene-derived SOA is provided by the online characterisation techniques coupled to CASC. We focus on ozonolysis in order to compare the results from CASC with a range of previous studies which measure SOA chemical composition (Bateman et al., 2009; Kundu et al., 2012; Maksymiuk et al., 2009; Walser et al., 2008). In addition, from a human health perspective, exposure to limonene SOA is most likely to occur indoors, where ozone is the dominant sink of limonene.

Before the introduction of reactants, the concentrations of $O_3$, $NO_x$ and particles in the clean chamber were below the detection limits of the respective instruments in Table 1. The relative humidity of the chamber air was adjusted to 40 % and 6 µL limonene (> 99%, Sigma) was added and mixed to produce a starting concentration of ~190 ppb based on PTR-MS

5    quantification. Ozone was introduced into the chamber over a 20 minute period, during which time the chamber air was regularly mixed; a maximum concentration of ~450 ppb was achieved (Figure 3). This corresponds to a stoichiometric excess of ozone with respect to the number of double bonds present in the limonene precursor. Ozonolysis was performed under dark conditions without the addition of $NO_x$.

SOA was produced rapidly following the introduction of ozone to the chamber (Figure 3). Particles grew via homogeneous nucleation into a single mode with diameter ~160 nm and standard deviation σ=0.21. The measured SMPS data (black curve) were corrected for particle wall losses (red curve) using a procedure similar to Rollins et al., (2009) which is

15    described in the supplementary information. Over 85% of the loss-corrected mass was formed within the first 30 minutes of ozone introduction, with slower additional growth over the next ~ 3 hours of the experiment.

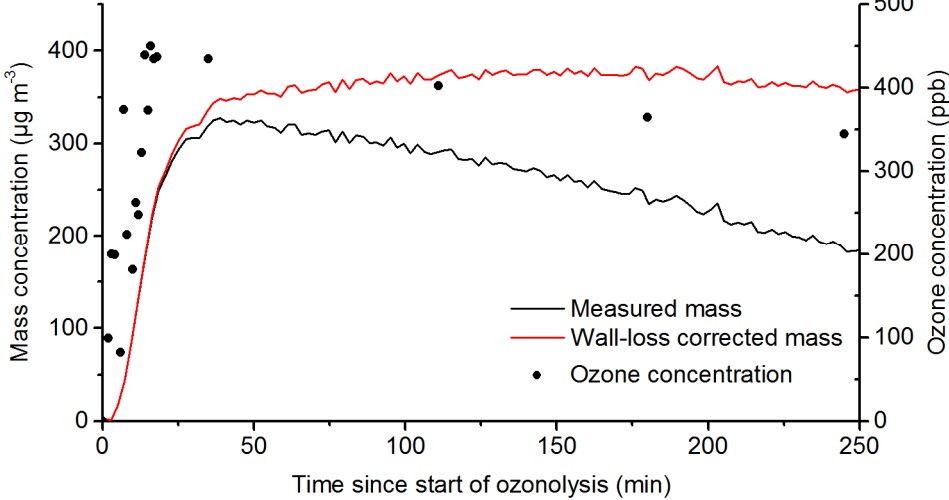

**Figure 3: Time series showing evolution of SOA mass (primary y-axis) and ozone concentration**
20    **(secondary y-axis) in the chamber. The measured SOA concentration (red curve) was corrected**
**to account for particle deposition to the chamber walls (black curve).**

## 3.1 Gas and particle molecular composition changes during ozonolysis

### 3.1.1 CASC measurements: PTR-MS and EESI-MS

Limonene was lost from the chamber over a period of 30 minutes, due to reaction with ozone
5    (Figure 4). Limonene was quantified using the PTR-MS from the sum of signals at *m/z* 137
([M + H]$^+$) and *m/z* 81 (major fragment). We detect a number of gas-phase products, noting
that structural isomers of the species described here cannot be distinguished in our analysis.
These were mostly assigned as [M + H]$^+$ and agree well with previous studies of limonene
and other terpene ozonolysis (Ishizuka et al., 2010; Lee et al., 2006). The largest yields are
10   for ubiquitous small acids and carbonyls such as formic acid (*m/z* 47, Figure 4), formaldehyde
(*m/z* 31, Figure 4), acetic acid, acetaldehyde and acetone. *m/z* 75 reported by Lee et al., (2006)
is present here, with an assigned neutral formula ($C_3H_6O_2$). Plausible structures require
secondary OH-mediated fragmentation of the limonene backbone and could include known
atmospheric consituents such as propanoic acid (Chebbi and Carlier, 1996), hydroxyacetone
15   (Zhou et al., 2009) or methyl acetate (Christensen et al., 2000).

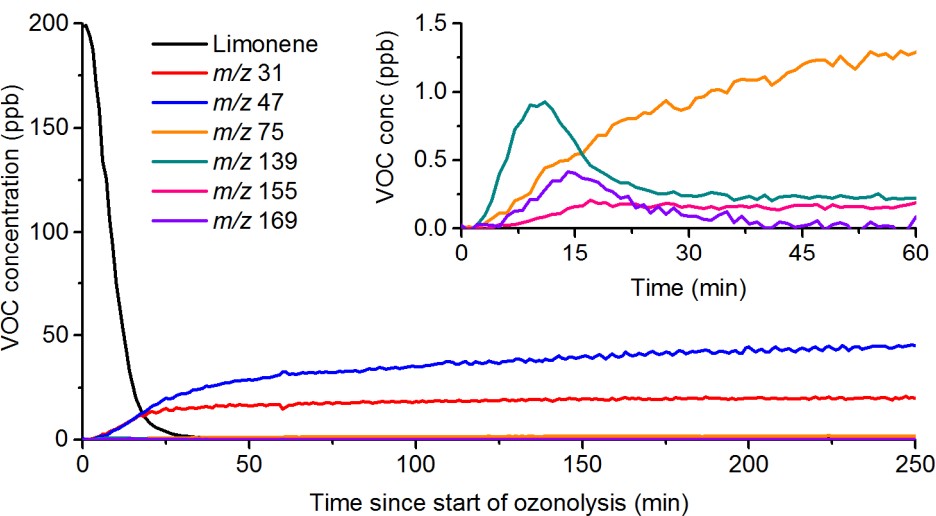

**Figure 4: Concentrations of selected gas-phase VOCs including limonene, formaldehyde (*m/z* 31)
and formic acid (*m/z* 47) detected using PTR-MS during limonene ozonolysis. The inset graph**

**shows products formed in lower concentrations, including first-generation unsaturated species (*m/z* 139, limonaketone and *m/z* 169, limonaldehyde) which are removed by further ozonolysis.**

Unsaturated gas-phase products corresponding to both reaction channels in Figure 1 can be detected at higher masses. In particular, limonaketone ($C_9H_{14}O$, *m/z* 139) is produced from the minor exo-double bond ozonolysis channel and is the most volatile unsaturated product (Donahue et al., 2007). It peaks in concentration at 11 minutes before being depleted, presumably by oxidation of the remaining double bond in the gas phase. Relatively few products from the ring-opening endo C=C channel can be quantified, but we note that limononaldehyde ($C_{10}H_{16}O_2$, *m/z* 169), the analogous ring-opening product to limonaketone, exhibits a similar time dependence and peaks at 14 minutes. Other products of endo C=C ozonolysis are generally condensable since a double functionalization reaction occurs without fragmenting the carbon backbone. Small gas phase signals for such products including *m/z* 155 ($C_9H_{14}O_2$) (Lee et al., 2006) (Figure 4), *m/z* 187 ($C_9H_{14}O_4$, limonic acid) and *m/z* 201 ($C_{10}H_{16}O_4$, 7-hydroxy limononic acid) were detected close to the instrument background. The same peaks were observed in limonene ozonolysis experiments performed by Ishisuka et al., (2010).

We discuss these products further in the context of particle-phase composition measurements from EESI-MS, which is applied to long term (> 4 hour) SOA monitoring for the first time here. An EESI mass spectrum 50 minutes after the start of ozonolysis is shown in Figure 5(a). As for the PTR measurements, isobaric compounds may complicate interpretation of the spectrum. The ion source was operated in negative ionisation mode and the most abundant [M − H]⁻ ions detected with EESI-MS compare well with major products identified in previous offline ESI-MS studies (Bateman et al., 2009; Kundu et al., 2012; Walser et al., 2008). We base potential assignments on previous literature. These include: *m/z* 185 (neutral formula $C_9H_{14}O_4$, limonic acid), *m/z* 171 ($C_8H_{12}O_4$, keto-limonalic acid), *m/z* 183 ($C_{10}H_{16}O_3$, limononic acid), *m/z* 199 ($C_{10}H_{16}O_4$, 7-hydroxy limononic acid). *m/z* 245 ($C_{11}H_{18}O_6$), one of the most abundant products from Kundu et al., (2012) is observed here and is along with other $C_{11}$-$C_{15}$ products is indicative of oligomerisation, specifically via the reactive uptake of gas-phase carbonyls to the particle phase.

Larger $C_{>20}$ products described by Kundu et al., (2012) and Bateman et al., (2009) could not be identified here. Since most products described in Kundu et al., (2012) are unsaturated, they would ultimately be oxidised in the conditions used here. Reactions of stabilised Criegee

intermediates with initial products could plausibly produce other high molecular weight species. However, these are not observed here and it is likely that the pre-concentration achieved by filter or impactor sampling in other studies leads to a greater sensitivity for species with very low concentrations compared to our online method.

We show time series for selected ions, along with the cumulative total ion current (TIC) across all ions, in Figure 5(b). The time resolution of the measurements is 4 minutes, which are shown as continuous lines for clarity. The EESI-MS ion source is stable over the > 4 hour time period of the experiment with respect both to the TIC and individual ion time series, demonstrating that the technique can operate continuously throughout long laboratory

experiments.

As demonstrated in Gallimore and Kalberer (2013) and Gallimore et al., (2017), relative intensity changes in EESI-MS can be used to infer relative concentration changes in the particle phase. Many individual time series scale approximately with the loss-corrected

particle mass (Figure 3) as shown in Figure 5(b). Slightly different upward (e.g. *m/z* 201, $C_9H_{14}O_5$) or downward (e.g. *m/z* 215, $C_{10}H_{16}O_5$) trends can be observed as aging continues on longer timescales. This suggests a slow production or loss from multi-generational chemistry either in the gas or particle phases, and is investigated in more detail later for *m/z* 201.

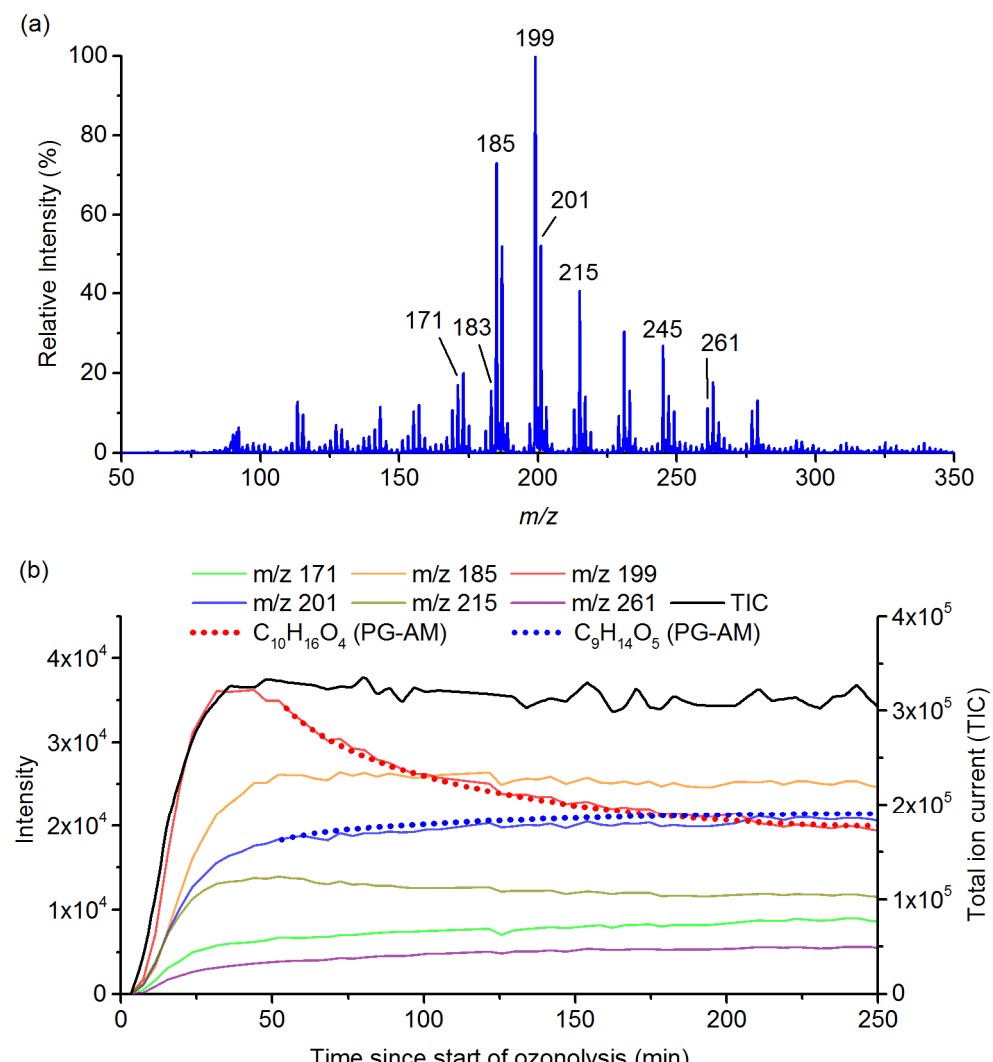

**Figure 5: (a) EESI mass spectrum after 50 minutes. Some major peaks discussed in the text are labelled with their *m/z*. (b) Wall-loss corrected intensities for selected particle-phase ions detected using EESI-MS during limonene SOA formation. The total ion current (TIC) across the entire MS is shown on the secondary y-axis. The dotted lines are a PG-AM simulation for the reaction of $C_{10}H_{16}O_4$ with ozone to form $C_9H_{14}O_5$, assuming semisolid SOA.**

A number of ions over a large molecular weight range (*m/z* 107-295) deviate significantly from the above trend and become depleted from the aerosol on longer timescales. We focus on the most abundant of these, *m/z* 199 (Figure 5(b)) which peaks after ~ 30 minutes and then

decays notably over the remainder of the experiment. Other ions showing similar trends are discussed later in section 3.2.

A few assignments exist for *m/z* 199 (neutral formula $C_{10}H_{16}O_4$), which has been observed
as a major product in previous studies (Kundu et al., 2012; Walser et al., 2008).  The most direct route to such an oxidised $C_{10}$ species involves ring-opening by ozonolysis of the endo C=C and hence preservation of the intact carbon backbone. For instance, 7-hydroxy limononic acid is a compelling assignment which is functionalised at the former endo C=C and still contains an intact exo C=C. Secondary formation routes may include reaction of
small carbonyls with initial $C_{<10}$ oxidation products in the particle phase, as discussed for *m/z* 245 above, or (unsaturated) hydroperoxide formation (Kundu et al., 2012).

We note that most plausible assignments for *m/z* 199 are unsaturated. The chemical loss of *m/z* 199 will therefore include ozonolysis reactions. In addition to SOA composition changes,
oxidation at longer times is also indicated by a continuing decrease in ozone after limonene has been consumed (Figure 3). We propose that this is via heterogeneous reaction due to the low volatility of any possible $C_{10}H_{16}O_4$ product. The very low corresponding gas phase signal (*m/z* 201, $[M + H]^+$) and previous observations of products which form only in the condensed phase (Maksymiuk et al., 2009) support this. Assuming $C_{10}H_{16}O_4$ to be 7-hydroxy-limononic
acid, a major ozonolysis product would be a carbonyl with formula $C_9H_{14}O_5$ (*m/z* 201, Figure S3). The *m/z* 201 time series in Figure 5(b) increases slowly during the period after 50 minutes when *m/z* 199 is depleted, consistent with this hypothesis. We explore this proposed reaction further using the PG-AM model below.Zhang et al., (2006) point out that for uptake coefficients ($\sim 10^{-3}$) typical of other model SOA systems such as oleic acid, the heterogeneous
loss rate of unsaturated species should be fast, and limited by their formation rate. By contrast, the time constant for loss of *m/z* 199 in this experiment is finite and relatively long, on the order of 1 hour. This is consistent with the study of Bateman et al., (2009), who found a $\sim 30$ minute lifetime for similar species (with higher ozone concentrations, $\sim 1$ ppm) using a time resolved analysis technique. Based on this slow loss, they estimated an effective
ozonolysis rate coefficient of $10^3$-$10^4$ $M^{-1}$ $s^{-1}$, consistent with some rate constants measured

in water (Hoigne and Bader, 1983) but 2-3 orders of magnitude lower than for a commonly used model SOA system, oleic acid (Lisitsyn et al., 2004).

We note here that other potentially unsaturated ions such as *m/z* 185 do not exhibit a decrease at longer times. Limonic acid is one likely assignment but Walser et al., (2008) have proposed saturated alternatives. It may be that the stable *m/z* 185 signal at longer times is a combination of loss of limonic acid and compensating production of other isobaric species, but we are unfortunately unable to investigate this further here.

### 3.1.2 PG-AM modelling of SOA composition changes

We investigate the time dependence of *m/z* 199 and 201 further, considering two possible scenarios: loss limited by bulk ozone-alkene reaction, and the formation of high viscosity particles which impede (an otherwise fast) bulk reaction. The latter of these has received substantial attention in recent years since the discovery that monoterpene-derived SOA can form an amorphous phase state (Virtanen et al., 2010). The characteristic time for bulk diffusion described in Shiraiwa et al., (2011) is on the order of minutes-hours for a semisolid accumulation mode particle, consistent with the 30-60 minutes estimated here and in Bateman et al., (2009).

The two scenarios were modelled by simulating reactive uptake of ozone to SOA containing unsaturated $C_{10}H_{16}O_4$ using the Pretty Good Aerosol Model (PG-AM, (Griffiths et al., 2009), (Gallimore et al., submitted)). The model was initialised to the experimental conditions 50 minutes after the introduction of ozone; an initial mole fraction of 0.05 was assumed for unsaturated $C_{10}H_{16}O_4$ based on the ion intensities in Figure 5(a) and further production was neglected. Physico-chemical model parameters were fixed (section 2.5), with the exception of the organic and oxidant diffusion coefficients, and condensed-phase ozonolysis rate constant, which were varied manually. Formation of the predicted carbonyl product of $C_{10}H_{16}O_4$ ozonolysis, $C_9H_{14}O_5$, was also simulated and the resulting reactant and product time series were compared to *m/z* 199 and 201 respectively in Figure 5(b).

In both scenarios, gas-phase loss of the reactant only made a very small contribution to the total loss rate, owing to its low vapour pressure and the relatively small gas-phase rate constant, $k_g^{II}$. In the bulk diffusion-limited scenario, a condensed-phase ozonolysis rate constant of $k_p^{II} = 10^6$ M$^{-1}$ s$^{-1}$, comparable to oleic acid, was assumed. To reproduce our observations, diffusion coefficients representative of semisolid SOA were required. The model simulation overlaid on Figure 5(b) is for $D_{org} = 10^{-16}$ and $D_{O3} = 5 \times 10^{-9}$ cm$^2$ s$^{-1}$ respectively. There is good correspondence between the model and measurements for both $C_{10}H_{16}O_4$ (m/z 199) and $C_9H_{14}O_5$ (m/z 201). This is consistent with the hypothesis that 7-hydroxy limononic acid and its carbonyl oxidation product make the dominant contribution to the measured m/z 199 and 201 signals. The model was found to be sensitive to both parameters and various combinations in the range $D_{org} = 10^{-15}$-$10^{-17}$ and $D_{O3} = 10^{-7}$-$10^{-9}$ cm$^2$ s$^{-1}$ provide reasonable fits to the data.

For the bulk reaction-limited scenario, the modelled concentration was sensitive only to the slow bulk ozonolysis rate constant, $k^{II}$, for any representative liquid SOA diffusion coefficients ($D_{O3} > 10^{-6}$ and $D_{org} > 10^{-10}$ cm$^2$ s$^{-1}$ respectively). $k^{II} = 8 \times 10^3$ M$^{-1}$ s$^{-1}$ was found to give the best agreement with our measurements, in good agreement with the range ($10^3$-$10^4$ M$^{-1}$ s$^{-1}$) estimated by Bateman et al., (2009). The modelled time series were very similar to the diffusion-limited case and so are omitted from Figure 5(b) for clarity.

The extent to which high particle viscosity influences reactivity is still an open question and we are not aware of any studies of the viscosity properties of limonene SOA. However, the diffusion coefficients used here are at the lower limit of what has been reported for monoterpene SOA at 40% RH (Renbaum-Wolff et al., 2013) ($D_{org} \geq 10^{-16}$ cm$^2$ s$^{-1}$) and significantly lower than some other determinations in SOA, e.g. (Hosny et al., 2016) ($D_{org} \sim 10^{-11}$ cm$^2$ s$^{-1}$). It therefore seems unlikely that the aerosol is sufficiently viscous to fully explain our data, but a combination of slow diffusion and reaction may well do, especially given further suggestions of inhibited reactivity discussed in the following section. A more detailed modelling investigation is out of the scope of this study given the lack of constraining experimental data (both here and in general for limonene SOA) and the assumptions made about the chemical identity of m/z 199 and 201. However, this prompts further study of the

diffusion characteristics of limonene SOA, especially since the SOA yield is higher and the aerosol components more oxidised than for many other biogenic VOCs.

**3.2 Particle-bound Reactive Oxygen Species (ROS) quantification**

**3.2.1 CASC measurements: OPROSI**

We also monitored the formation of particle-bound reactive oxygen species (ROS) during the chamber experiment. The raw ROS concentration data were wall loss corrected using the same procedure as for the particle mass, and the corrected data are presented in Figure 6. We report ROS quantities as an equivalent concentration of hydrogen peroxide per cubic metre

of air which reflects the effective reactivity of the ROS present in the aerosol to the assay (red curve in Figure 6). The appearance of ROS is highly correlated in time with the formation of SOA mass in the chamber (Figure 3) within the first few minutes of ozone introduction. However, while the SOA mass concentration continues to increase slowly for a number of hours, the ROS signal reaches a small maximum after around 30 minutes. Both tend towards

relatively stable values after the first hour of oxidative chemistry in the chamber, consistent with the relatively small composition changes in the aerosol (Figure 5(b)). To make this more explicit, we normalise this measured quantity to the mass of SOA present to give a relative in-particle concentration (blue curve in Figure 6). The mass-weighted ROS concentration is highest in the early stages of the reaction, before tending to a stable value of $0.42 \pm 0.04$ nmol

$[H_2O_2]$ $\mu g^{-1}$ after the first hour of the experiment.

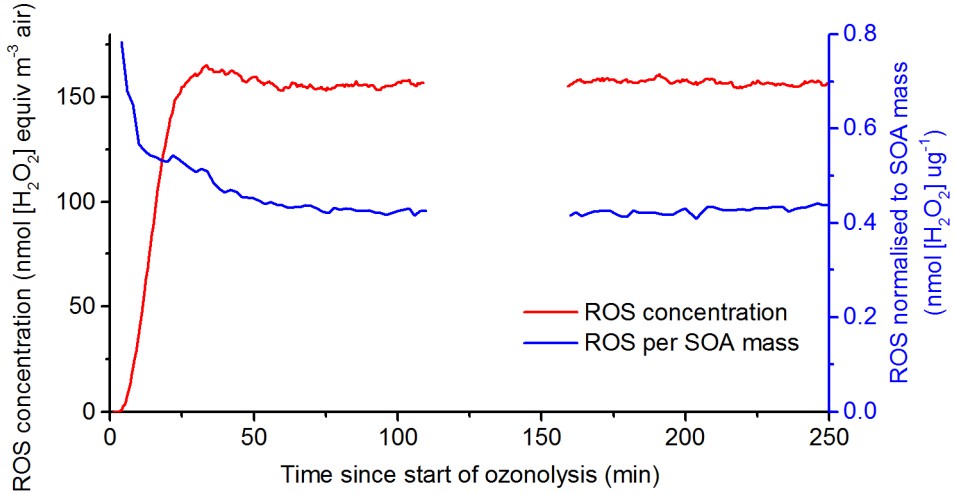

**Figure 6: Wall loss corrected particle-bound ROS detected using OPROSI during limonene SOA formation. The procedural blank at 110-160 minutes has been removed for clarity. The secondary y-axis shows an effective ROS concentration per mass of aerosol, which is highest at the start of the experiment and reaches a stable value of 0.42 ± 0.04 nmol [H₂O₂] per µg SOA.**

We present an analysis of the above time series which proposes that the total ROS signal can be split in to two components with different characteristic lifetimes:

$$[ROS_{total}] = [ROS_{long}] + [ROS_{short}] \qquad (1)$$

We propose that $ROS_{long}$ are a group of relatively stable long-lived products (such as hydrogen peroxide and organic peroxides) which constitute the stable ROS at the end of the experiment and which have been shown to be major products of monoterpene ozonolysis (Docherty et al., 2005; Wang et al., 2011). Meanwhile $ROS_{short}$ are reactive species (possibly radicals or otherwise short-lived compounds such as reactive peroxides) which are produced directly from ozonolysis or other early-generation reactions. Previous studies have concluded that a substantial fraction of ROS present in laboratory (Fuller et al., 2014) and ambient (Huang et al., 2016) aerosols is short lived.

If we assume that [ROS$_{long}$] scales with the total particle mass in proportion to the final mass-weighted ROS concentration (as do most individual aerosol components in Figure 5(b)), the net contribution of [ROS$_{short}$] to the total measured signal can be estimated:

5    $$[ROS_{total}] = 0.42 \times Mass_{SOA} + [ROS_{short}] \quad (2)$$

Figure 7(a) shows the estimated contributions of long- and short-lived ROS to the total measured ROS signal using Equation 2. Based on this simple analysis, long-lived ROS makes the dominant contribution to the total ROS signal over the course of the experiment. As

10   expected from the mass-weighted ROS curve in Figure 6, short-lived ROS is most important early in the reaction when reactive species are being produced by ozonolysis from the limonene precursor (Figure 4). If some ROS$_{short}$ were converted to ROS$_{long}$ during the early part of the experiment, Equation 2 could underestimate the ROS$_{short}$ contribution to [ROS$_{total}$] and correspondingly overestimate [ROS$_{long}$] early in the experiment.

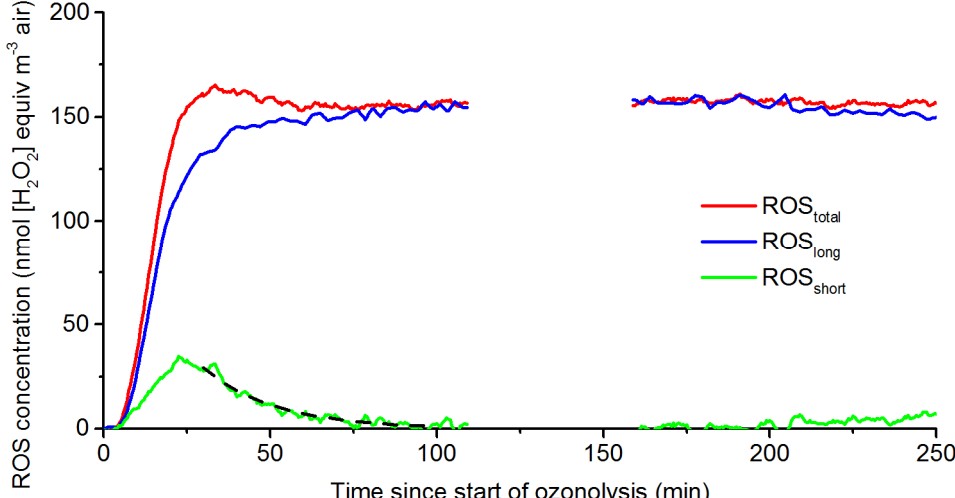

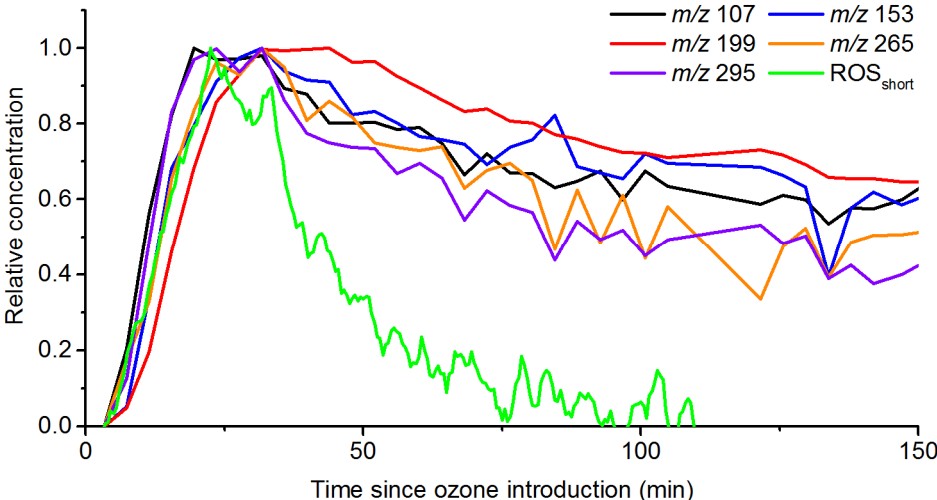

**Figure 7: (a) Total particle-bound ROS decomposed into short-lived and long-lived components according to Equation 2. The black dashed line represents an exponential fit to ROS$_{short}$ with a characteristic lifetime of 17 minutes. (b) Relative abundance of a number of particle-phase ions which decay substantially over the course of the experiment. It is possible that some of these are tracers for ROS$_{short}$ (green curve) if isobaric species interfere with the measurements.**

[ROS$_{short}$] reaches a maximum 23 minutes into the experiment and tends towards zero at longer times. This reflects the competition between production and loss for short-lived species. The apparent production rate of ROS$_{short}$ will depend on two factors. Firstly, chemical production (likely predominantly in the gas phase, but also via heterogeneous ozonolysis of later generation products), which will be fastest at the start of ozonolysis when precursor concentrations are at their highest. Secondly, gas-particle partitioning of reactive species, which will increasingly favour the particle phase as the mass loading in the chamber increases. A convolution of these factors is a plausible explanation for the maximum in [ROS$_{short}$] in Figure 7(a). At longer times, where the concentration of organic radical precursors (e.g. limonene) is reduced, chemical loss of ROS$_{short}$ dominates and by definition in Equation 2 tends to zero at longer times.

We estimate the lifetime of particle-bound ROS$_{short}$ by neglecting further production of short-lived species after 30 minutes, since the rate of change of chemical composition (Figures 4

and 5) and particle mass (Figure 3) has slowed by this point. We note that the 4 minute time resolution of the instrument (determined mainly by mixing times in the particle collector and fluorescence detector cell) will act as a lower limit on the apparent $ROS_{short}$ lifetime in this analysis. The black dashed curve in Figure 7(a) represents an exponential fit to the data

between 30 and 110 minutes and yields a characteristic time for pseudo first-order $ROS_{short}$ loss of 17 minutes. This may be a overestimate if $ROS_{short}$ production is not negligible during this time. Equation 2 also neglects any possible secondary chemistry involved in forming long-lived ROS.

We have interrogated the EESI-MS data for species which may act as tracers for $ROS_{short}$ based on their time dependence. As mentioned above, in addition to *m/z* 199, a number of other ions (over the range *m/z* 107-295) show a characteristic maximum at early times in the reaction followed by notable decay over the remainder of the experiment (Figure 7(b)). All of these ions could contain enough oxygen atoms to possess ROS-active functional groups,

although we do not obtain definitive molecular formulae with low resolution EESI-MS and a lack of previous literature assignments. While none of these ions map perfectly onto the $ROS_{short}$ time series, several species may be present for each low resolution peak such that the loss of some ROS compounds is obscured by others with different time dependences. An alternative explanation is that, like *m/z* 199, at least a partial contribution to the signals may

be unsaturated and therefore oxidised over time. Despite the general difficulty of detecting peroxides and other related ROS species via MS, recent progress has been made in this area (Steimer et al., 2017). More definitive detection of ROS and its precursors (e.g. condensed unsaturated species) via EESI-MS is a direction for future work.

**3.2.2 Comparison to other studies of ROS formation in SOA**

Chen and Hopke (2010), Chen et al., (2011) and Chen et al., (2017) studied ROS formation from the ozonolysis of limonene using a similar chemical assay with an offline filter sampling and sonication and filter extraction method. Like the current study, both short-lived and long-lived ROS are reported. However, $ROS_{long}$ yields reported by Chen and Hopke (2010) and

Chen et al., (2017) (0.15-0.19 nmol $[H_2O_2]$ $\mu g^{-1}$) were lower than those determined here (0.42

nmol [$H_2O_2$] $\mu g^{-1}$). A number of experimental differences may be important. The three other studies employed dry conditions, compared to 40% RH here. The presence of water may influence the gas-phase fate of initial products and promote ROS formation (for instance, hydroperoxides from reaction of stabilised Criegee intermediates with water (Docherty et al., 2005)) as well as potentially modifying Henry's law partitioning of species such as hydrogen peroxide, and facilitating oligomerisation and hydrolysis reactions in the condensed phase (Gallimore et al., 2011). The higher mass loading here (375 $\mu g$ $m^{-3}$) compared to these previous studies (30-160 $\mu g$ $m^{-3}$) may be an important parameter through its influence on gas-particle partitioning and subsequent particle-phase reaction.

Chen et al., (2011) reported a correlation between [$O_3$]/[VOC] and [$ROS_{long}$] for a range of VOCs, and found higher ROS yields when ozone was in excess, presumably as a result of increased formation of oxygenated products such as peroxides. This is consistent with the higher [$ROS_{long}$] reported here ([$O_3$]$_{max}$/[limonene]$_0$ = 2.4) compared to Chen et al., (2017) ([$O_3$]$_0$/[limonene]$_0$ = 0.45). Furthermore, we proposed above that oxidation of the second (exo) double bond is partly occurring in the particle phase; this direct ROS formation in the particle may result in higher measured yields than gas phase only routes. These findings contrast with Chen and Hopke (2010) who do not see a systematic trend in [$ROS_{long}$] with varying [$O_3$]/[limonene].

Qualitatively, the ROS observations in the current study compare well with Fuller et al., (2014) who noted that oxidised oleic acid aerosols contained both short- and long-lived ROS components using the same assay. The quantitative differences are summarised in Table 2. In summary, limonene ROS was formed in overall higher yield and the short-lived components were in general longer lived. These differences can be rationalised by considering the molecular properties of oleic acid and limonene.

| | Limonene SOA (this study) | Oleic acid SOA (Fuller et al., 2014) |
|---|---|---|
| $ROS_{long}$ yield (nmol [$H_2O_2$] $\mu g^{-1}$) | 0.42 | 0.14 |
| Maximum fraction of $ROS_{short}$ (%) | 25-40 | 75 |
| $ROS_{short}$ lifetime (mins) | 17 | a few |

**Table 2: Quantitative differences in ROS formation between the current study (limonene SOA) and Fuller et al., (2014) (oleic acid SOA). More long-lived ROS is formed for limonene, but short-lived ROS appears more important for oleic acid.**

The majority (~75 %) of ROS detected for freshly oxidised oleic acid particles was short-lived in nature, compared to a relatively smaller fraction for limonene SOA in the current study. Oleic acid is of low volatility and subject only to heterogeneous ozonolysis in the particle, so ROS production will be in-situ in the particle phase. As discussed above, the oxidation of limonene is a more complex multiphase process, and at least initially a large

fraction of organic intermediates will be produced in the gas-phase. Since such species have short gas-phase lifetimes, many will go on to react further (e.g. to make long-lived ROS by reaction with water as discussed above (Docherty et al., 2005)) before entering the particle phase, leading to an overall different ratio of short- and long-lived ROS in the oleic acid and limonene-SOA particles.

The overall yield of $ROS_{long}$ is higher for limonene by a factor of 3. Limonene contains two C=C, and therefore more potential to form highly oxygenated products including ROS than oleic acid with a single C=C. The fates of initial products may also be different as a result both of the reaction phase (purely particle phase vs multiphase) and the presence of water

vapour in the current experiments which favours hydroperoxide formation (Docherty et al., 2005).

Finally, the $ROS_{short}$ lifetime in Fuller et al., (2014) was shorter (a few minutes) than that reported here (17 minutes). We note that the oleic acid aerosol samples were collected onto

filters, stored at room temperature for different lengths of time and extracted into solution for offline analysis, which decoupled $ROS_{short}$ production and loss. This is not possible for online sampling and as discussed above this could result in an overestimated $ROS_{short}$ lifetime for limonene SOA. Particle viscosity may also play a role: oleic acid remains liquid throughout ozonolysis (Hosny et al., 2016) and so in-particle reactions which consume $ROS_{short}$ should

not be inhibited. If limonene SOA were viscous, as considered above, this could extend the effective lifetime of short lived species.

The very recent study of Tuet et al., (2017) also found that the oxidative potential of SOA, quantified using a different assay, depends on the hydrocarbon precursor. Particularly high redox activity was found for naphthalene-derived SOA. However, comparing directly between different assays is challenging because the relative sensitivity towards different organic components (e.g. peroxides, quinones, radicals, polyaromatic hydrocarbons) is not well established (Fang et al., 2015).

## 4 Conclusions

The new 5.4 m$^3$ Cambridge Atmospheric Simulation Chamber (CASC) facility enables atmospheric chemical processes to be studied in the laboratory under relevant conditions with a high degree of time and chemical resolution. The characteristics of the chamber in terms of lights, mixing and wall losses have been thoroughly characterised as an important benchmark for current and future studies. The multiphase oxidation of limonene was studied using a range of continuous time-resolved particle and gas-phase measurements. The merits of highly time resolved measurements of particle composition and reactive oxygen species (ROS) were demonstrated and the links between the two explored for limonene SOA.

The majority of particle bound ROS detected in limonene SOA is long lived on the experiment timescale (4 hours) suggesting an important role for such health-relevant species in ambient particles. The overall yield of ROS was also significantly higher than for another SOA model system studied with the same methodology, oleic acid. This may have important implications for indoor air quality in particular given the abundance of limonene in cleaning and "air freshening" products. Even for relatively simple model SOA systems, the time-dependent characteristics of ROS are variable and reflect the underlying chemistry of the gas and particle phases in terms of reactivity, partitioning and viscosity.

The apparently slow loss of unsaturated species via heterogeneous ozonolysis, and the relatively long decay time of ROS$_{short}$, both provide indirect evidence of a role for viscous particle formation in limonene SOA. We note that substantial uncertainties remain associated with in-particle diffusion and gas-particle exchange in viscous organic aerosols. However, if

particle viscosity is impeding chemical reactivity, these particles are in essence a reservoir for reactive organics (both unsaturated and health-relevant). Such reactive carbon may therefore enjoy an extended lifetime in the atmosphere before reacting in more humid regions, or in particular the elevated RH and temperature conditions of the human airways.

The potential atmospheric and health implications of this hypothesis merit further study for limonene SOA and other aerosol systems.

## Supplementary information

Chamber characterisation data relating to the spectra of the light sources, the mixing of VOCs, and particle wall loss rates are provided. A proposed reaction scheme for particle-

phase oxidation of unsaturated species is also included.

## Acknowledgements

This work was funded by the European Research Council (grant 279405), the UK Natural Environment Research Council (grant NE/H52449X/1) and the Velux foundation (project number 593).

## Data access

Data presented in this study can be obtained by contacting the corresponding author.

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
