# Peer review of "Multiphase composition changes and reactive oxygen species formation during limonene oxidation in the new Cambridge Atmospheric Simulation Chamber (CASC)"

_Atmospheric Chemistry and Physics, 2017_

## Referee Comment (RC2)

Review for "Multiphase composition changes and reactive oxygen species formation during limonene oxidation in the new Cambridge Atmospheric Simulation Chamber (CASC)" by Peter J. Gallimore et al.

General comments: this manuscript showed interesting results about the online composition changes of gas and particle phase products during the photolysis of limonene by using mass spectrometry. Meanwhile, they also measured the reactive oxygen species (ROS) formation by limonene SOA in water by using a fluorescent assay. Based on these experiments and mathematic modelling, the authors claimed that diffusion-limited and bulk reaction-limited scenarios might have resulted in the low loss of some low volatile compounds like 7-hydroxy limononic acid ($C_{10}H_{16}O_4$). Furthermore, the authors also claimed that stable ROS dominate the total ROS formed by limonene SOA in water especially in a long timescale during the oxidation of limonene in the Cambridge Atmospheric Simulation Chamber (CASC). Overall the results are interesting and the manuscript was written well. If my following concerns can be addressed, I would like to recommend this manuscript to be published in Atmos. Chem. Phys.

Specific points:

1. The title of "Multiphase composition changes and reactive oxygen species formation during limonene oxidation in the new Cambridge Atmospheric Simulation Chamber (CASC)" shows that the ROS in this article was generated during the limonene oxidation in CASC. However, the ROS data in Fig. 6 and 7 were relevant to the limonene SOA dissolved water solutions by using Online Particle-bound Reactive Oxygen Species Instrument (OPROSI). Even though some kind of ROS (organic peroxides etc.) could be generated during the limonene SOA formation process, the title is not accurate to describe the source of the ROS in this article.

2. In line 16-18 of page 2: "Similarly, organic reactive oxygen species (ROS), including organic peroxides and oxygen centred radicals, are thought to be associated with the observed negative health effects of airborne particles (Verma et al., 2009)." The authors introduced the definition of ROS for the first time in this article. However, they did not clarify the

difference of the term ROS used in this article from that in literatures (e.g. Klaus Apel and Heribert Hirt., Annu. Rev. Plant Biol. REACTIVE OXYGEN SPECIES: Metabolism, Oxidative Stress, and Signal Transduction. 55, 373-399, 2004; Josep M. Anglada et al., Interconnection of Reactive Oxygen Species Chemistry across the Interfaces of Atmospheric, Environmental, and Biological Processes. Acc. Chem. Res. 48, 575-583, 2015.), especially the authors should clarify the ROS species their method (OPROSI) could characterize.

3. In Fig. 6 at page 18, the author showed a plateau of ROS formation in limonene SOA water solutions (0.42 nmol $[H_2O_2]$ $\mu g^{-1}$). Afterwards, the authors used the equations 1 and 2 (page 19) to categorize the total ROS to short and long modes. During this analysis, the assumption of "$[ROS_{long}]$ scales with the total particle mass in proportion to the final mass weighted ROS concentration (as do most individual aerosol components in Figure 5(b))….." has been used. However, the plateau in Fig. 6 may be induced by a homeostasis of long and short lifetime ROS. So the used equivalence of $[ROS_{long}]=0.42\times MASS_{SOA}$ can overestimate the yield of $ROS_{long}$. In the same timescale, the yield of limonene SOA is also relatively stable (Fig.3), so it is reasonable to see the plateau of EESI mass spectrum intensity in Fig. 5(b). If the authors would like to connect the plateau of Fig. 5(b) with the plateau Fig. 6, they need a response sensitivity test to confirm the ROS value indicated by the OPROSI system are real relevant to the ions showed in Fig. 5.

4. In line 6-10: "We propose that $ROS_{long}$ are a group of relatively stable long-lived products (such as organic peroxides) which constitute the stable ROS at the end of the experiment, and $ROS_{short}$ are reactive species (possibly radicals or otherwise short-lived compounds such as reactive peroxides) species which are produced directly from ozonolysis or other early-generation reactions." The authors should discuss more about the component of $ROS_{long}$ and $ROS_{short}$. In addition, numerous studies indicated that limonene SOA and other precursor-generated SOA particles could show high oxidative potential and generate ROS, like: Chen, X., and Hopke, P. K.: A chamber study of secondary organic aerosol formation by limonene ozonolysis, Indoor air, 20, 320-328, 2010.; Wang, Y., Kim, H., and Paulson, S. E.: Hydrogen peroxide generation from α-and β-pinene and toluene secondary organic aerosols, Atmospheric environment, 45, 3149-3156, 2011.; McWhinney, R. D., Zhou, S., and Abbatt, J. P. D.: Naphthalene SOA: redox activity and naphthoquinone gas–particle partitioning, Atmos. Chem. Phys., 13, 9731-9744, 10.5194/acp-13-9731-2013, 2013.;

Badali, K. M., Zhou, S., Aljawhary, D., Antiñolo, M., Chen, W. J., Lok, A., Mungall, E., Wong, J. P. S., Zhao, R., and Abbatt, J. P. D.: Formation of hydroxyl radicals from photolysis of secondary organic aerosol material, Atmos. Chem. Phys., 15, 7831-7840, 2015.; Tong, H., Arangio, A., Lakey, P., Berkemeier, T., Liu, F., Kampf, C., Pöschl, U., and Shiraiwa, M.: Hydroxyl radicals from secondary organic aerosol decomposition in water, Atmos. Chem. Phys., 16, 1761-1771, 2016. Tuet, W. Y., Chen, Y., Xu, L., Fok, S., Gao, D., Weber, R. J., and Ng, N. L.: Chemical oxidative potential of secondary organic aerosol (SOA) generated from the photooxidation of biogenic and anthropogenic volatile organic compounds, Atmospheric Chemistry and Physics, 17, 839-853, 2017.

5. In 2010, Chen and Hopke have measured the ROS formation by limonene SOA (Chen, X., and Hopke, P. K., Indoor air, 20, 320-328, 2010.) using a similar fluorescent assay system. Their study showed a maximum ~0.2 nmol $[H_2O_2]$ $\mu g^{-1}$. However, current study showed a yield of 0.2 nmol $[H_2O_2]$ $\mu g^{-1}$, which is 2 times higher. More recently, they also found that when limonene SOA mass concentration ranged from 30.3 to 157.3 $\mu g\ m^{-3}$, the ROS concentration could range from 6.1 to 29.4 nmol $m^{-3}$ of $H_2O_2$ (Chen, et al., Aerosol and Air Quality Research, 17, 59-68, 2017.), this value is also much lower than the value of ~150 nmol $m^{-3}$ in Fig. 6. How to explain this?

6. In 2014, Epstein et al. indicated that photolysis can influence the abundance of peroxide in biogenic SOA (Environ. Sci. Technol., 48, 11251-11258, 2014.). The authors are encouraged to discuss the potential impact of the photolysis on their ROS values.

7. Some typos should be corrected: page 5: line 3 "1/4" and 1/2"", line 17 and 18:"160W","75W". Page 9: line 15: "4mm".

---

## Referee Comment (RC1) · Anonymous Referee #1 · 10 Apr 2017

**Multiphase composition and reactive oxygen species formation during limonene oxidation in the new Cambridge Atmsopheric Simulation Chamber (CASC) by Gallimore et al.**

**Reviewer comments**

This is an interesting paper highlighting the capabilities of a new simulation chamber in Cambridge. A variety of online measurement techniques were used to characterise the gas and particles formed during limonene ozonolysis. Of particular interest are the online reactive oxygen species measurements, showing potential difference in the times scale for ROS formation. I do have a number of concerns that need to be clarified before publication in ACP.

**General comments**

*Experimental*
Firstly there is too little experimental detail given in this paper. The authors direct the reader to other papers for basic details on the mass spectrometers. For instance the flow rate and mass analyser (and indeed the mass resolution) used for EESI is not given. Since this is a unique instrument, the reader should be given much more extensive details of the instrumentation and its capabilities without needing to read another paper alongside this one. There are lots of cases where the reader is directed to a paper that has been submitted and so I cannot judge the links inferred.

Again, since this is the first chamber paper I need more details. The chamber is apparently collapsible but I couldn't work out if this was what was happening or was a dilution flow being used? What was the final volume of the chamber and does that impact wall losses? There are lots of details of the lamps and then the NOx chemistry, but then I assume these are not actually used in the one experiment that is shown? There seems to be a disconnect- is this a chamber characterisation paper (which is limited) or a SOA characterisation paper? Most of the chamber characterisation is in the SI.

*Diffusion versus ozone uptake*
Firstly more details are needed about the model. Is partitioning based solely on equilibrium partitioning and if so how were the vapour pressures of the products determined? How was the reaction rate coefficient of ozone with the products determined? I would have thought a C10 species with only 4 oxygens would be a semi-volatile species and so its profile could be impacted by its gas phase reactivity as well, with subsequent re-volatilisation. However, I cannot tell from the data presented how the model deals with this.

Clearly m/z 199 shows a different profile than the other species shown. However, this is not the only ion shown with a double bond. *m/z 185* is most likely limonic acid ($C_9H_{13}O_4$). This also has an intact double bond but clearly does not show the same effect. Have you looked for any other species with an intact double bond? Can you predict what the product of m/z 199 might be and look for the trend in that? I realise it may be complicated by isobaric species.

*ROS quantification*

I have a concern here about the method used to correct the data. Was the ROS and/or the SOA mass corrected for particle loss? On reading its seems like you use the ROS measured in the chamber and divide this by the loss corrected particle mass (I have assumed this is what you have done). If this is the case, I disagree with his approach. The ROS you have measured is based on what is actually in the chamber when you measure. The amount of SOA mass is much lower than the corrected number. Thus you are normalising to particle mass that is not present. If you used the actual measured particle mass the trend would look very different, increasing at longer reaction times. This needs to be clarified and the approach justified.

**Specific points**

In general the text is well written and easy to follow.

Page 1: not sure you need "new" in the title
Page 3, line 3: Give estimate of limonene emission
Page 3: There is very little given here about previous studies of limonene. I would expect some more background.
Page 4, line 1: Change to "was studied"
Page 4, line 3: FEP given before explained
Page 5, Fig 2: Collapsible spelt wrong. Im assuming there is no dilution here. What is the mechanism that allows the chamber to collapse?
Page 6, line 5: How clean is the zero air? Any peaks in PTR-MS above detection limit?
Page 6, line 9: Im surprised you don't see any OVOC from the water. How often is it changed?
Page 7: As described above there is far too little experimental detail included here, especially for the ROS and EESI-MS. How many OVOC standards have you investigated to ensure there is no in-source dimers formed or in-source fragmentation? Ive looked at the Gallimore and Kalberer paper, but there is very limited information on using the signal as a pseudo-quantification. Do you think the changing mix of organics will lead to any matrix effects?
Page 8, lines 3-4: Need spaces between units
Page 8, line 11: change to "can be associated"
Page 8, line 21: I assume this should be "Stainless steel'. Was a filter used in the PTR-MS sample line?
Page 10, line 1: a-pinene is a rather volatile species to use to account for wall losses. Please justify its use here.
Page 10, line 16: This section lacks details rather than relying on a different paper.
Page 11: I was rather surprised after the characterisation section that only 1 experiment was included. How representative are the results here of other ozone – limonene experiments? Why not show a OH reaction as well?
Page 11, line 14: give ± 1$\sigma$ on diameter

Page 12, line 6: You use the term "characteristic" but I don't know what this applies to? It sounds like a description of more than one experiment but that is not presented here. Page 12, line 6: insert "the PTR"

Page 12, line 14: Which of these structures is most likely based on mechanisms.

Page 12, Fig 4: The purple and blue lines are very similar. Can an ozone profile be included for comparison.

Page 12, line 8: Limonaldehyde appears to form slightly later that the limonaketone. How do these compare to the ROS short profile?

Page 13, line 12: Can these species be seen in previous studies using PTR-MS. I don't know but Im surprised you don't see them at all.

Page 13, line 26: Do you think that dimers are present based on the masses observed? I would think even if both double bonds are oxidised you would still see species up to C18, say from reaction of the stabilised Criegee intermediate with other products.

Page 14, line 12: can you estimate the elemental composition of these ions?

Page 14, line 19: I got a bit confused as to how small carbonyls were related? Do you mean heterogeneous or in-particle chemistry of two smaller OVOC is forming a C10 compound rather than the first stages of limonene oxidation?

Page 19, line 2: I don't like the use of the word decomposed – suggests some chemistry. Perhaps use "split"

Page 20, fig 7: Can you predict possible elemental formulae for the small ions? How efficient is gas phase removal of OVOC products in the ROS injection system?

Page 23, line 13: I don't understand what is meant by "collected in an offline manner". Needs some more details.

**SI**

Table legends need to be above the tables.

Page 3, Table S1: can you add what kind of lamps are in the other chambers for comaprsion.

Page 5, line 7: Were the particles dried or not for the wall loss experiments?

Page 6, line 13: How does this yield compare to previous studies?

---

## Author Comment (AC1) · 14 Jun 2017

**Author response – Reviewer #1**

Reviewer comments
*This is an interesting paper highlighting the capabilities of a new simulation chamber in Cambridge. A variety of online measurement techniques were used to characterise the gas and particles formed during limonene ozonolysis. Of particular interest are the online reactive oxygen species measurements, showing potential difference in the times scale for ROS formation. I do have a number of concerns that need to be clarified before publication in ACP.*

We thank the reviewer for these constructive comments and address them point-by-point below.

General comments

Experimental
*Firstly there is too little experimental detail given in this paper. The authors direct the reader to other papers for basic details on the mass spectrometers. For instance the flow rate and mass analyser (and indeed the mass resolution) used for EESI is not given. Since this is a unique instrument, the reader should be given much more extensive details of the instrumentation and its capabilities without needing to read another paper alongside this one. There are lots of cases where the reader is directed to a paper that has been submitted and so I cannot judge the links inferred.*

We have described the unique capabilities and operating conditions of the EESI-MS instrument in substantially more detail and subdivided the Chamber Instrumentation section to improve clarity. The submitted paper (Gallimore et al., 2017) is now published and cited in relevant parts of this manuscript.

The new section 2.2.1 (p8 line 10–p9 line 19) now reads:

"2.2.1 Extractive Electrospray Ionisation Mass Spectrometry

Extractive Electrospray Ionisation Mass Spectrometery (EESI-MS) is an online particle analysis technique and the design and optimisation of our EESI source is described in Gallimore and Kalberer (2013). It contains a commercial electrospray probe (Thermo Scientific HESI-II) with a custom-built aerosol injector and manifold. The primary electrospray was operated at a voltage of –3.0 kV and a $N_2$ sheath flow rate of 0.8 L $min^{-1}$. A water–methanol 1:1 mixture (Optima LC-MS grade solvents; Fisher Scientific) containing 0.05% formic acid (90%, Breckland Scientific) was infused into the ESI probe at 10 µL $min^{-1}$. Chamber air was delivered into the source at 1 L $min^{-1}$. Collision and extraction of the SOA particles by primary electrospray droplets occurs in an off-axis configuration with respect to the MS inlet, to minimise source contamination and memory effects through particle deposition. The EESI source was coupled to an ion-trap mass spectrometer (Thermo Scientific LTQ Velos). Spectra were acquired in the negative ionisation mode over the range *m/z* 50-500, with a mass resolution ~ 2000 (full width at half maximum, FWHM) at *m/z* 400.

Gallimore and Kalberer (2013) demonstrated that the relative EESI-MS ion intensity correlated with the mass concentration of tartaric acid particles delivered into the source, suggesting that the entire particle bulk is extracted for analysis. More recently, Gallimore et al., (2017) showed that the kinetics of particle-phase reactions could be monitored; loss rates derived from EESI-MS measurements compared well with other studies, and spectra were compared to Liquid Chromatography (LC) MS to confirm that the EESI-MS assignments were present in the aerosol rather than formed as artefacts in the ion source."

The following sub-headings have been added, and the OPROSI section expanded:

P9, line 26: "2.2.2. Proton Transfer Reaction Mass Spectrometry"

P10, line 9: "2.2.3 Online Particle-bound Reactive Oxygen Species Instrument"

P10, lines 13–25: "The continuous sample inflow (5 L min$^{-1}$) passes through a PM$_{2.5}$ cyclone (URG-2000-30E-5-2.5-S) and charcoal denuder prior to entering into a particle-into-liquid sampler (PILS). Particles are collected into a 1 mL min$^{-1}$ spray containing horseradish peroxidase (HRP) (TypeVI, 1 unit mL$^{-1}$ in 10% phosphate buffer solution (PBS), Sigma Aldrich) which reacts with ROS present in the particles. This is combined with a 1mL min$^{-1}$ aqueous 2'7'-dichlorofluorescein (DCFH) solution (10µM, 10% PBS, Sigma Aldrich), which is oxidised to a fluorescent product (DCF) by the ROS-HRP solution. After a 10 minute reaction time at 40 °C the concentration of DCF is quantified via fluorescence spectroscopy. The fluorescence response is calibrated with H$_2$O$_2$ and quantitative ROS concentrations are reported as "[H$_2$O$_2$] equivalents". The assay also responds to organic peroxides. It is likely sensitive to HO$_x$ radicals and ions such as superoxide but we are unable to obtain suitable standards to test this directly."

More general discussion of EESI-MS (previously p7 lines 4-9) has been moved from the Methods section to the Introduction (p4 line 14–p5 line 1): "EESI retains the key advantage of "soft" electrospray ionisation MS techniques, namely that quasi-molecular ions are produced from aerosol analytes with minimal fragmentation. Individual molecular species can be identified and relative intensity changes monitored over time as a measure of concentration changes with particles (Gallimore et al., 2017)."

*Again, since this is the first chamber paper I need more details. The chamber is apparently collapsible but I couldn't work out if this was what was happening or was a dilution flow being used? What was the final volume of the chamber and does that impact wall losses? There are lots of details of the lamps and then the NOx chemistry, but then I assume these are not actually used in the one experiment that is shown? There seems to be a disconnect- is this a chamber characterisation paper (which is limited) or a SOA characterisation paper? Most of the chamber characterisation is in the SI.*

We have added clarification regarding chamber sampling, which does not use a dilution flow (p7 lines 25–28): "Typically a maximum of 1.5 m$^3$ air was removed during a batch experiment, and guide rails on the chamber frame allow the bag to

inflate and deflate at ambient pressure. Up to ~60 % of the chamber volume could be sampled using this system if required."

We have added an addition sentence to section 2.4 (p12 lines 23–24): "The potential impact of changes in chamber volume during sampling (maximum ~30 %) on wall loss rates was not considered."

We performed a thorough set of characterisation measurements, including on the light sources and resulting $NO_x$ photo-stationary state, before embarking on SOA experiments. We felt it was important to benchmark the chamber so that future studies are informed by these measurements and the reader can compare CASC to other chambers. The SOA discussion which follows, although performed with the lights off, illustrates the instrumental and scientific capabilities of the chamber and uses characterisation measurements such as particle wall loss rates.

We have added the following clarification (p14 lines 18-19): "Ozonolysis was performed under dark conditions without the addition of $NO_x$."

**Diffusion versus ozone uptake**
*Firstly more details are needed about the model. Is partitioning based solely on equilibrium partitioning and if so how were the vapour pressures of the products determined? How was the reaction rate coefficient of ozone with the products determined? I would have thought a C10 species with only 4 oxygens would be a semi-volatile species and so its profile could be impacted by its gas phase reactivity as well, with subsequent re-volatilisation. However, I cannot tell from the data presented how the model deals with this.*

We have added additional details to this section which more fully describe partitioning and diffusion in the model. For the simulations presented in the discussion manuscript, we assumed the $C_{10}H_{16}O_4$ product was non-volatile. In response to the reviewer's question we calculated a partitioning coefficient and included a gas-phase loss for this species. Based on later comments about potential products of this reaction we have included such a product ($C_9H_{14}O_5$) in the model which can be compared to *m/z* 201.

The revised and extended section reads (p12 line 26–p14 line 2): "Illustrative model simulations were performed using the Pretty Good Aerosol Model (PG-AM). PG-AM is described in detail in Griffiths et al., (2009) and Gallimore et al., (2017). The model treats the following processes in a kinetic framework: chemical reaction in both the gas and particle phases, gas-particle exchange via uptake and evaporation, and diffusion within the particle. Fluxes between the gas and particle phases depend on the aerosol surface area as well as each species' accommodation coefficient ($\alpha$, dimensionless) and partitioning coefficient ($K$, M atm$^{-1}$). Diffusion is parameterised according to Griffiths et al., (2009); the particle is treated as a series of nested shells, with the rate of transport of each species between shells determined by its diffusion coefficient ($D$, cm$^2$ s$^{-1}$). The differential equations governing reaction and diffusion are integrated forwards in time using Mathematica (v11, Wolfram)."

In this study, reaction of unsaturated $C_{10}H_{16}O_4$ with ozone was simulated for a single particle of characteristic radius $r_{eff} = 3V_t/S_t = 84$ nm based on the measured total particle volume ($V_t$) and surface area ($S_t$). $C_{10}H_{16}O_4$ was assumed to be 7-hydroxy limononic acid for vapour pressure calculations and formation of a corresponding carbonyl oxidation product, $C_9H_{14}O_5$, was represented in the model mechanism (Figure S3). The vapour pressures of $C_{10}H_{16}O_4$ and $C_9H_{14}O_5$ were estimated at ~2.8 × $10^{-5}$ Pa and ~2.2 × $10^{-6}$ Pa using the online model EVAPORATION (Compernolle et al., 2011) which performed well in our tests for a species with known vapour pressure, pinic acid. The resulting partitioning coefficients calculated from these vapour pressures, $K \sim 2 \times 10^{10}$ M atm$^{-1}$ and $\sim 2 \times 10^{11}$ M atm$^{-1}$ means that both species are almost entirely condensed (< 99 %) under the experimental conditions.

The accommodation coefficient for organic species was fixed at $\alpha_{org} = 0.1$. Ozone partitioning ($K_{O3} = 0.1$ M atm$^{-1}$) (Morris et al., 2002) and accommodation ($\alpha_{O3} = 10^{-3}$) (Gallimore et al., 2017) were fixed based on the literature for oleic acid particles. The particle phase bimolecular rate constant for ozonolysis ($k_p^{II}$) and the ozone and organic diffusion coefficients ($D_{O3}$ and $D_{org}$) were varied as described in the results. A fixed gas-phase rate constant for exo double bond ozonolysis, $k_g^{II} = 7 \times 10^{-18}$ cm$^3$ molecule$^{-1}$ s$^{-1}$ (Zhang et al., 2006) was also included. However, gas-phase loss was not competitive with particle phase oxidation due to this relatively small rate constant and the low vapour pressure of $C_{10}H_{16}O_4$. The carbonyl product $C_9H_{14}O_5$ was formed with a yield of 0.4, based on the branching of exo-C=C ozonolysis products for β-pinene (Jenkin, 2004). We did not attempt to account for the fate of the other reaction branch featuring a $C_{10}$ Criegee intermediate due to the wide range of possible Criegee intermediate products in the condensed phase including peroxides, carbonyls and secondary ozonides (Lee et al., 2012; Maksymiuk et al., 2009)."

The "submitted" paper (Gallimore et al., 2017), which includes a full model description and characterisation also is now published and referred to here.

*Clearly m/z 199 shows a different profile than the other species shown. However, this is not the only ion shown with a double bond. m/z 185 is most likely limonic acid (C9H13O4). This also has an intact double bond but clearly does not show the same effect. Have you looked for any other species with an intact double bond? Can you predict what the product of m/z 199 might be and look for the trend in that? I realise it may be complicated by isobaric species.*

We agree that *m/z* 185 and other ions may be unsaturated, and add the following discussion (p20 line 28–p21 line 2): "We note here that other potentially unsaturated ions such as *m/z* 185 do not exhibit a decrease at longer times. Limonic acid is one likely assignment but Walser et al., (2008) have proposed saturated alternatives. It may be that the stable *m/z* 185 signal at longer times is a combination of loss of limonic acid and compensating production of other isobaric species, but we are unfortunately unable to investigate this further here."

We show a number of time series with a similar time dependence over a wide *m/z* range in Figure 7 which may plausibly be unsaturated. We do not wish to speculate on their specific structures without supporting evidence but add the following sentence

(Line p27 lines 3–5): "An alternative explanation is that, like $m/z$ 199, at least a partial contribution to the signals may be unsaturated and therefore oxidised over time."

Following the reviewer's suggestion, we have investigated possible products of $m/z$ 199 ozonolysis and find the most likely particle-phase product to have $m/z$ 201. We have added discussion (p20 lines 11–15): "Assuming $C_{10}H_{16}O_4$ to be 7-hydroxy-limononic acid, a major ozonolysis product would be a carbonyl with formula $C_9H_{14}O_5$ ($m/z$ 201, Figure S3). The $m/z$ 201 time series in **Error! Reference source not found.**(b) increases slowly during the period after 50 minutes when $m/z$ 199 is depleted, consistent with this hypothesis. We explore this proposed reaction further using the PG-AM model below."

This proposed reaction is illustrated in the supporting information (Figure S3).

Additional discussion is provided in the PG-AM section (p21 lines 21–23): "Formation of the predicted carbonyl product of $C_{10}H_{16}O_4$ ozonolysis, $C_9H_{14}O_5$, was also simulated and the resulting reactant and product time series were compared to $m/z$ 199 and 201 respectively in **Error! Reference source not found.**(b)."

This continues (p22 lines 1–4): "There is good correspondence between the model and measurements for both $C_{10}H_{16}O_4$ ($m/z$ 199) and $C_9H_{14}O_5$ ($m/z$ 201). This is consistent with the hypothesis that 7-hydroxy limononic acid and its carbonyl oxidation product make the dominant contribution to the measured $m/z$ 199 and 201 signals."

A product time series has been added to an updated Figure 5(b), and the caption has been updated accordingly (p19 lines 5–6): "The dotted lines are a PG-AM simulations for the heterogeneous reaction of $C_{10}H_{16}O_4$ with ozone to form $C_9H_{14}O_5$, assuming semisolid SOA."

Given this additional discussion we have subdivided the molecular composition section (3.1) into PTR-MS and EESI-MS measurements from CASC (3.1.1, p15 line 7) and PG-AM modelling (3.1.2).

ROS quantification

*I have a concern here about the method used to correct the data. Was the ROS and/or the SOA mass corrected for particle loss? On reading its seems like you use the ROS measured in the chamber and divide this by the loss corrected particle mass (I have assumed this is what you have done). If this is the case, I disagree with his approach. The ROS you have measured is based on what is actually in the chamber when you measure. The amount of SOA mass is much lower than the corrected number. Thus you are normalising to particle mass that is not present. If you used the actual measured particle mass the trend would look very different, increasing at longer reaction times. This needs to be clarified and the approach justified.*

Apologies for the confusion. Both the particle mass (SMPS) and particle-bound ROS signal (OPROSI) were loss corrected using the same procedure. Before correction, both raw signals decrease at longer times due to wall losses. We have added the following clarification (p23 lines 4–5): "The raw ROS concentration data were wall

loss corrected using the same procedure as for the particle mass, and the corrected data are presented in **Error! Reference source not found.**."

*In general the text is well written and easy to follow.*

*Page 1: not sure you need "new" in the title*

Since this is the first characterisation and demonstration of CASC, we would like to keep "new" in the title.

*Page 3, line 3: Give estimate of limonene emission*

The sentence now reads (p3 lines 3–4): "Limonene is one of the most abundant BVOCs in the troposphere, with an estimated biogenic emission rate of 11 Tg yr$^{-1}$ (Guenther et al., 2012)."

*Page 3: There is very little given here about previous studies of limonene. I would expect some more background.*

We have included some extra background on the chemistry of limonene SOA. The revised background reads (p3 line 13–p4 line 3): "Limonene contains two reactive C=C double bonds which results in multiple generations of oxidation products (Bateman et al., 2009; Kundu et al., 2012; Walser et al., 2008) containing a range of functional groups including carboxylic acids, carbonyls, peroxides and alcohols. Previous studies have mainly focused on the reaction of limonene with ozone (Kundu et al., 2012; Zhang et al., 2006), with relatively few OH-aging experiments reported, particularly with respect to chemical characterisation (Zhao et al., 2015). Ozone is a major sink for limonene under a range of atmospheric conditions (Atkinson and Arey, 2003) and will dominate in indoor scenarios which may be most relevant for the health effects of limonene SOA (Waring, 2016). The endo C=C of limonene is more susceptible to ozonolysis by a factor of 10-50 (Zhang et al., 2006) and some of the first-generation ring opening products are condensable (**Error! Reference source not found.**). Subsequent oxidation of the remaining double bond may therefore occur in either the gas or condensed phases depending on the properties of the initial products and the aerosol loading.

The ability of limonene to form multifunctional products via successive oxidation steps results in high aerosol yields relative to other terpenes (Hoffmann et al., 1997; Zhang et al., 2006). Aside from ozonolysis, other condensed-phase reactions further modify the composition of limonene SOA. Kundu et al., (2012) report the reactive uptake of carbonyls to form oligomeric products, while the formation of light-absorbing "brown carbon" via uptake and reaction of ammonia and amines appears to be particularly efficient for limonene SOA compared to other precursors (Bones et al., 2010; Updyke et al., 2012)."

*Page 4, line 1: Change to "was studied"*

This has been changed (p4 line 9).

*Page 4, line 3: FEP given before explained*

We have changed "FEP" to "Teflon" (p4 line 11).

*Page 5, Fig 2: Collapsible spelt wrong. Im assuming there is no dilution here. What is the mechanism that allows the chamber to collapse?*

The spelling has been corrected (Figure 2). The modified experimental section discussed above now describes the guide rails which allow the bag to inflate and deflate without dilution (p7 lines 25–28).

*Page 6, line 5: How clean is the zero air? Any peaks in PTR-MS above detection limit?*

We have modified the following sentence (p7 lines 2–4): "The chamber is filled with air from a zero air generator (KA-MT2, Parker Hannifin, UK) which uses a molecular sieve, an activated charcoal bed and filters to remove water, VOCs and particulates respectively." See response to the comment below for contaminant quantification via PTR-MS.

*Page 6, line 9: Im surprised you don't see any OVOC from the water. How often is it changed?*

The water is changed every week at minimum, as now detailed (p7 line 7): "…which is replaced at least each week."

On revisiting the detailed PTR-MS data we could observe OVOC introduction – apologies for this earlier error. We have amended the discussion accordingly (p7 lines 8–10): "We monitored changes in particle and VOC concentrations during water introduction and found particle formation to be negligible, but observed up to ~10 ppb acetone and lesser quantities (< 2 ppb) of $C_1$-$C_2$ aldehydes and acids."

*Page 7: As described above there is far too little experimental detail included here, especially for the ROS and EESI-MS. How many OVOC standards have you investigated to ensure there is no in-source dimers formed or in-source fragmentation? Ive looked at the Gallimore and Kalberer paper, but there is very limited information on using the signal as a pseudo-quantification. Do you think the changing mix of organics will lead to any matrix effects?*

As discussed in the general section, we have created sub-sections and included substantially more experimental detail for EESI-MS (p8 line 11–p9 line 19) and OPROSI (p10 lines 10–28).

The EESI-MS section cites the now-published Gallimore et al., (2017) and emphasises that our EESI-MS spectra compare well with LC-MS (suggesting ions correspond to aerosol analytes rather than in-source artefacts) and can be used for kinetic experiments (suggesting that relative quantification is not significantly influenced by evolving SOA composition).

*Page 8, lines 3-4: Need spaces between units*

We have introduced spaces (p10 lines 1–2).

*Page 8, line 11: change to "can be associated"*

This has been changed (p10 line 10).

*Page 8, line 21: I assume this should be "Stainless steel'. Was a filter used in the PTR-MS sample line?*

We have corrected this to "Stainless steel" (p11 line 2). The PTR-MS sample line was not filtered.

*Page 10, line 1: a-pinene is a rather volatile species to use to account for wall losses. Please justify its use here.*

We have deleted reference to α-pinene wall losses in this section.

*Page 10, line 16: This section lacks details rather than relying on a different paper.*

As discussed in the general comments above we have revised and extended this section extensively (p12 line 26–p14 line 2).

*Page 11: I was rather surprised after the characterisation section that only 1 experiment was included. How representative are the results here of other ozone – limonene experiments? Why not show a OH reaction as well?*

The experiment shown following the characterisation section was intended to be illustrative of the capabilities of the chamber during SOA experiments, particularly the unique EESI-MS and OPROSI instruments. We compare to other studies of limonene SOA formation and find good agreement, for example in terms of oxidation products identified, but also make new contributions, for example in terms of the time dependence and yield of ROS from SOA.

We presented an ozonolysis experiment for two reasons: Firstly, ozone is a major sink for limonene under all conditions, especially indoors. Secondly, previous studies focus on ozonolysis, with few OH studies reported, and comparison of results from our new chamber with previous studies was an important aspect of this paper. We emphasised these points in the introduction discussed above (p3 lines 17–22) and add an additional sentence here (p14 lines 5–9): "We focus on ozonolysis in order to compare the results from CASC with a range of previous studies which measure SOA chemical composition (Bateman et al., 2009; Kundu et al., 2012; Maksymiuk et al., 2009; Walser et al., 2008). In addition, from a human health perspective, exposure to limonene SOA is most likely to occur indoors, where ozone is the dominant sink of limonene."

*Page 11, line 14: give ± 1σ on diameter*

This now reads (p14 lines 22–23): "…a single mode with diameter ~160 nm and standard deviation σ=0.21."

*Page 12, line 6: You use the term "characteristic" but I don't know what this applies to? It sounds like a description of more than one experiment but that is not presented here.*

We have rephrased this sentence to read (p9 lines 9–10): "Limonene was lost from the chamber over a period of 30 minutes, due to reaction with ozone (**Error! Reference source not found.**)."

*Page 12, line 6: insert "the PTR"*

We have corrected this (p15 line 11).

*Page 12, line 14: Which of these structures is most likely based on mechanisms.*

All the possible structures require multiple secondary reactions and we are not in a position to assess which if any are most likely. We have added the following discussion (p15 line 19–p16 line 2): "Plausible structures require secondary OH-mediated fragmentation of the limonene backbone and could include known atmospheric consituents such as propanoic acid (Chebbi and Carlier, 1996), hydroxyacetone (Zhou et al., 2009) or methyl acetate (Christensen et al., 2000)."

*Page 12, Fig 4: The purple and blue lines are very similar. Can an ozone profile be included for comparison.*

Adding an ozone profile to Fig 4 would result in a very crowded plot. We have therefore added the profile to Fig 3 (p4 line 1) and updated the caption accordingly (p4 lines 2–5): "Figure 3: Time series showing evolution of SOA mass (primary y-axis) and ozone concentration (secondary y-axis) in the chamber. The measured SOA concentration (red curve) was corrected to account for particle deposition to the chamber walls (black curve)."

We have also added discussion of the ozone profile in the context of particle-phase ozonolysis (p20 lines 3–5): "In addition to SOA composition changes, oxidation at longer times is also indicated by a continuing decrease in ozone after limonene has been consumed (**Error! Reference source not found.**)."

*Page 12, line 8: Limonaldehyde appears to form slightly later that the limonaketone. How do these compare to the ROS short profile?*

The reviewer is correct; limononaldehyde peaks at 14 minutes, limonaketone at 11. We have noted the limononaldehyde time in the text (p16 line 17).

The shape of these species resembles $ROS_{short}$, which peaks later at 23 minutes (added, p26 line 3). Although possibly tracers for short-lived ROS, the OPROSI should not respond to gas-phase VOCs due to the charcoal denuder on the instrument's inlet.

*Page 13, line 12: Can these species be seen in previous studies using PTR-MS. I don't know but Im surprised you don't see them at all.*

These peaks were present in the PTR-MS study of Ishizuka et al., (2010). We added a note to this effect (p17 lines 1–2): "The same peaks were observed in limonene ozonolysis experiments performed by Ishisuka et al., (2010)."

*Page 13, line 26: Do you think that dimers are present based on the masses observed? I would think even if both double bonds are oxidised you would still see species up to C18, say from reaction of the stabilised Criegee intermediate with other products.*

We agree that dimers could form even following the oxidation of both C=C, and we see evidence of oligomerisation reactions which are now mentioned (p17 lines 14–17): "*m/z* 245 ($C_{11}H_{18}O_6$), one of the most abundant products from Kundu et al., (2012) is observed here and is along with other $C_{11}$-$C_{15}$ products is indicative of oligomerisation, specifically via the reactive uptake of gas-phase carbonyls to the particle phase."

We have added a sentence discussing possible stabilised CI oligomers (p17 lines 20–22): "Reactions of stabilised Criegee intermediates with initial products could plausibly produce other high molecular weight species. However, these are not observed here and it is likely that…"

*Page 14, line 12: can you estimate the elemental composition of these ions?*

We now discuss *m/z* 201 ($C_9H_{14}O_5$) as a likely oxidation product of *m/z* 199 (p20 lines 11–15, response to general comment above) and have added an estimated composition for *m/z* 215 ($C_{10}H_{16}O_5$, p18 line 6).

*Page 15, line 19: I got a bit confused as to how small carbonyls were related? Do you mean heterogeneous or in-particle chemistry of two smaller OVOC is forming a C10 compound rather than the first stages of limonene oxidation?*

That is correct. We have added the following clarification (p19 line 20–p20 line 1): "Secondary formation routes may include reaction of small carbonyls with initial $C_{<10}$ oxidation products in the particle phase, as discussed for *m/z* 245 above…"

*Page 19, line 2: I don't like the use of the word decomposed – suggests some chemistry. Perhaps use "split".*

We have used "split" as suggested (p6 line 7).

*Page 20, fig 7: Can you predict possible elemental formulae for the small ions? How efficient is gas phase removal of OVOC products in the ROS injection system?*

For the major products discussed above our assignments are based on previous literature reports. Since the small ions discussed here (m/z 107 and 153) have not been previously proposed as products we are reluctant to speculate. P26 line 29–p27 line 1 now reads: "All of these ions could contain enough oxygen atoms to possess ROS-active functional groups, although we do not obtain definitive molecular formulae with low resolution EESI-MS and a lack of previous assignment in the literature."

We emphasise that major products are assigned based on previous literature (p17 line 11): "We base potential assignments on previous literature."

We employ a charcoal denuder to remove VOCs from the gas phase before ROS analysis. The denuder removes ozone to below detection limit levels and while we have

not tested organic standards, such denuders are routinely used in aerosol sampling to remove VOCs (Grover et al., 2005; Salvador and Chou, 2014).

*Page 23, line 9: I don't understand what is meant by "collected in an offline manner". Needs some more details.*

We have clarified this point (p29 lines 10–12): "We note that the oleic acid aerosol samples were collected onto filters, stored at room temperature for different lengths of time and extracted into solution for offline analysis, which decoupled $ROS_{short}$ production and loss."

SI
*Table legends need to be above the tables.*

The table legends have been moved.

*Page 3, Table S1: can you add what kind of lamps are in the other chambers for comaprsion.*

The light sources for each chamber have been added (p3, Table S1).

*Page 5, line 7: Were the particles dried or not for the wall loss experiments?*

We have deleted "optionally" (p5, line 7) and added the following sentence (p5 lines 9–10): "No significant dependence on chamber RH was found for the range tested (0-70 %)."

*Page 6, line 13: How does this yield compare to previous studies?*

We have added the following comparison (p6 lines 13–16): "Aerosol yields are dependent on a number of parameters including aerosol mass loading and oxidation conditions, as well as chamber-specific factors. Our yields are consistent with previous studies at similar mass loadings, which occupy a broad range from ~30-90 % (Chen and Hopke, 2010; Leungsakul et al., 2005; Youssefi and Waring, 2014; Zhang et al., 2006)."

**References**

[revised manuscript text omitted]

---

## Author Comment (AC2) · 14 Jun 2017

**Author response – Reviewer #2:**

General comments
*This manuscript showed interesting results about the online composition changes of gas and particle phase products during the photolysis of limonene by using mass spectrometry. Meanwhile, they also measured the reactive oxygen species (ROS) formation by limonene SOA in water by using a fluorescent assay. Based on these experiments and mathematic modelling, the authors claimed that diffusion-limited and bulk reaction-limited scenarios might have resulted in the low loss of some low volatile compounds like 7-hydroxy limononic acid (C10H16O4). Furthermore, the authors also claimed that stable ROS dominate the total ROS formed by limonene SOA in water especially in a long timescale during the oxidation of limonene in the Cambridge Atmospheric Simulation Chamber (CASC). Overall the results are interesting and the manuscript was written well. If my following concerns can be addressed, I would like to recommend this manuscript to be published in Atmos. Chem. Phys.*

We thank the reviewer for these comments and respond point-by-point below.

Specific points:
*1. The title of "Multiphase composition changes and reactive oxygen species formation during limonene oxidation in the new Cambridge Atmospheric Simulation Chamber (CASC)" shows that the ROS in this article was generated during the limonene oxidation in CASC. However, the ROS data in Fig. 6 and 7 were relevant to the limonene SOA dissolved water solutions by using Online Particle-bound Reactive Oxygen Species Instrument (OPROSI). Even though some kind of ROS (organic peroxides etc.) could be generated during the limonene SOA formation process, the title is not accurate to describe the source of the ROS in this article.*

The reviewer points out that the detected ROS may be a combination of ROS formed in the aerosol, and additional ROS formed in aqueous solution following particle collection. While recent studies have indicated that OH radicals form when SOA is dissolved in water (Badali et al., 2015; Tong et al., 2016) the proposed mechanisms involve light or transition metals, neither of which are present in the OPROSI collection system.

Organic peroxides and other related functional groups have long been shown to be a major component of monoterpene SOA (Camredon et al., 2007; Docherty et al., 2005) and in Wragg et al., (2016) we demonstrated that the assay used in our study is sensitive to organic peroxide standards. We therefore prefer to keep the title as is, and address some of the specific points about different types of ROS in response to comment 2.

*2. In line 16-18 of page 2: "Similarly, organic reactive oxygen species (ROS), including organic peroxides and oxygen centred radicals, are thought to be associated with the observed negative health effects of airborne particles (Verma et al., 2009)." The authors introduced the definition of ROS for the first time in this article. However, they did not clarify the difference of the term ROS used in this article from that in literatures (e.g. Klaus Apel and*

*Heribert Hirt., Annu. Rev. Plant Biol. REACTIVE OXYGEN SPECIES: Metabolism, Oxidative Stress, and Signal Transduction. 55, 373-399, 2004; Josep M. Anglada et al., Interconnection of Reactive Oxygen Species Chemistry across the Interfaces of Atmospheric, Environmental, and Biological Processes. Acc. Chem. Res. 48, 575-583, 2015.), especially the authors should clarify the ROS species their method (OPROSI) could characterize.*

We have modified the introduction to clarify this definition of ROS (p2 lines 16–22): "Similarly, species including hydrogen peroxide and oxygen-centred radicals and ions can cause biological stress and damage (Anglada et al., 2015; Apel and Hirt, 2004). Related organic compounds including peroxides have been shown to be major SOA components (Camredon et al., 2007; Docherty et al., 2005). Together, these reactive oxygen species (ROS) are thought to be associated with the observed negative health effects of airborne particles (Verma et al., 2009)."

An OPROSI experimental subsection (2.2.3) has been added which describes the species characterised by OPROSI (p10 lines 21–25): "The fluorescence response is calibrated with $H_2O_2$ and quantitative ROS concentrations are reported as "[$H_2O_2$] equivalents". The assay also responds to organic peroxide standards. It is likely sensitive to $HO_x$ radicals and ions such as superoxide but we are unable to obtain suitable standards to test this directly."

*3. In Fig. 6 at page 18, the author showed a plateau of ROS formation in limonene SOA water solutions (0.42 nmol [H2O2] μg -1 ). Afterwards, the authors used the equations 1 and 2 (page 19) to categorize the total ROS to short and long modes. During this analysis, the assumption of "[ROSlong] scales with the total particle mass in proportion to the final mass weighted ROS concentration (as do most individual aerosol components in Figure 5(b))….." has been used. However, the plateau in Fig. 6 may be induced by a homeostasis of long and short lifetime ROS. So the used equivalence of [ROSlong]=0.42×MASSSOA can overestimate the yield of ROSlong. In the same timescale, the yield of limonene SOA is also relatively stable (Fig.3), so it is reasonable to see the plateau of EESI mass spectrum intensity in Fig. 5(b). If the authors would like to connect the plateau of Fig. 5(b) with the plateau Fig. 6, they need a response sensitivity test to confirm the ROS value indicated by the OPROSI system are real relevant to the ions showed in Fig. 5.*

By definition in Equation 2, the final $ROS_{long}$ yield will be 0.42 nmol [$H_2O_2$] μg$^{-1}$. We have added discussion of possible overestimated $ROS_{long}$ yields at the start of the experiment (p24 line 31–p25 line 2): "If some $ROS_{short}$ were converted to $ROS_{long}$ during the early part of the experiment, Equation 2 could underestimate the $ROS_{short}$ contribution to [$ROS_{total}$] and correspondingly overestimate [$ROS_{long}$] early in the experiment."

Regarding connecting the "plateaus" in Fig 5(b) and Fig 6, we are not proposing a direct link between the specific ions in Fig 5(b) and ROS, but using this as an illustration that the general aerosol composition (and gas phase composition and particle mass) is not changing significantly in the ROS plateau region.

*4. In line 6-10: "We propose that ROSlong are a group of relatively stable long-lived products (such as organic peroxides) which constitute the stable ROS at the end of the experiment, and ROSshort are reactive species (possibly radicals or otherwise short-lived compounds such as reactive peroxides) species which are produced directly from ozonolysis or other early-generation reactions." The authors should discuss more about the component of ROSlong and ROSshort. In addition, numerous studies indicated that limonene SOA and other precursor-generated SOA particles could show high oxidative potential and generate ROS, like: Chen, X., and Hopke, P. K.: A chamber study of secondary organic aerosol formation by limonene ozonolysis, Indoor air, 20, 320-328, 2010.; Wang, Y., Kim, H., and Paulson, S. E.: Hydrogen peroxide generation from α-and β-pinene and toluene secondary organic aerosols, Atmospheric environment, 45, 3149-3156, 2011.; McWhinney, R. D., Zhou, S., and Abbatt, J. P. D.: Naphthalene SOA: redox activity and naphthoquinone gas–particle partitioning, Atmos. Chem. Phys., 13, 9731-9744, 10.5194/acp-13-9731-2013, 2013.; Badali, K. M., Zhou, S., Aljawhary, D., Antiñolo, M., Chen, W. J., Lok, A., Mungall, E., Wong, J. P. S., Zhao, R., and Abbatt, J. P. D.: Formation of hydroxyl radicals from photolysis of secondary organic aerosol material, Atmos. Chem. Phys., 15, 7831-7840, 2015.; Tong, H., Arangio, A., Lakey, P., Berkemeier, T., Liu, F., Kampf, C., Pöschl, U., and Shiraiwa, M.: Hydroxyl radicals from secondary organic aerosol decomposition in water, Atmos. Chem. Phys., 16, 1761-1771, 2016. Tuet, W. Y., Chen, Y., Xu, L., Fok, S., Gao, D., Weber, R. J., and Ng, N. L.: Chemical oxidative potential of secondary organic aerosol (SOA) generated from the photooxidation of biogenic and anthropogenic volatile organic compounds, Atmospheric Chemistry and Physics, 17, 839-853, 2017.*

We have added the following additional detail for $ROS_{long}$ (p24 lines 11–14): "We propose that $ROS_{long}$ are a group of relatively stable long-lived products (such as hydrogen peroxide and organic peroxides) which constitute the stable ROS at the end of the experiment and which have been shown to be major products of monoterpene ozonolysis (Docherty et al., 2005; Wang et al., 2011)."

We are not in a position to speculate more on the identity of $ROS_{short}$ than we already do (p24 line 15): "…radicals or otherwise short-lived compounds such as reactive peroxides…") because unlike the long lived components where surrogate standards are available, we are unsure of the relative reactivity of the OPROSI assay towards different short-lived species.

We have cited some of the suggested references throughout the manuscript and we discuss Chen and Hopke (2010) in more detail below.

*5. In 2010, Chen and Hopke have measured the ROS formation by limonene SOA (Chen, X., and Hopke, P. K., Indoor air, 20, 320-328, 2010.) using a similar fluorescent assay system. Their study showed a maximum ~0.2 nmol [H2O2] μg -1 . However, current study showed a yield of 0.4 nmol [H2O2] μg -1 , which is 2 times higher. More recently, they also found that when limonene SOA mass concentration ranged from 30.3 to 157.3 μg m-3 , the ROS concentration could range from 6.1 to 29.4 nmol m-3 of H2O2 (Chen, et al., Aerosol and Air Quality Research, 17, 59-68, 2017.), this value is also much lower than the value of ~150 nmol m-3 in Fig. 6. How to explain this?*

Thank you for bringing these relevant studies to our attention. We discuss and explain differences between the studies (p27 line 10–p28 line 5): "Chen and Hopke (2010), Chen et al., (2011) and Chen et al., (2017) studied ROS formation from the ozonolysis of limonene using a similar chemical assay with an offline sampling and sonication extraction method. Like the current study, both short-lived and long-lived ROS are reported. However, $ROS_{long}$ yields for Chen and Hopke (2010) and Chen et al., (2017) (0.15-0.19 nmol [$H_2O_2$] $\mu g^{-1}$) were lower than those determined here (0.42 nmol [$H_2O_2$] $\mu g^{-1}$). A number of experimental differences may be important. The three other studies employed dry conditions, compared to 40% RH here. The presence of water may influence the gas-phase fate of initial products and promote ROS formation (for instance, hydroperoxides from reaction of stabilised Criegee intermediates with water (Docherty et al., 2005)) as well as potentially modifying Henry's law partitioning of species such as hydrogen peroxide, and facilitating oligomerisation and hydrolysis reactions in the condensed phase (Gallimore et al., 2011). The higher mass loading here (375 $\mu g$ $m^{-3}$) compared to these previous studies (30-160 $\mu g$ $m^{-3}$) may be an important parameter through its influence on gas-particle partitioning and subsequent particle-phase reaction.

Chen et al., (2011) reported a correlation between [$O_3$]/[VOC] and [$ROS_{long}$] for a range of VOCs, and found higher ROS yields when ozone was in excess, presumably as a result of increased formation of oxygenated products such as peroxides. This is consistent with the higher [$ROS_{long}$] reported here ([$O_3$]$_{max}$/[limonene]$_0$ = 2.4) compared to Chen et al., (2017) ([$O_3$]$_0$/[limonene]$_0$ = 0.45). Furthermore, we proposed above that oxidation of the second (exo) double bond is partly occurring in the particle phase; this direct ROS formation in the particle may result in higher measured yields than gas phase only routes. These findings contrast with Chen and Hopke (2010) who do not see a systematic trend in [$ROS_{long}$] with varying [$O_3$]/[limonene]."

We subdivided the ROS discussion into CASC measurements (3.2.1, p23 line 2) and comparison with other studies (3.2.2, p27 line 10) to improve readability of this extended ROS section.

*6. In 2014, Epstein et al. indicated that photolysis can influence the abundance of peroxide in biogenic SOA (Environ. Sci. Technol., 48, 11251-11258, 2014.). The authors are encouraged to discuss the potential impact of the photolysis on their ROS values.*

Photolysis is clearly an important fate for peroxides and ROS-relevant species, as indicated by Epstein et al., (2014) and other references from point 4. However, the ROS data in Figure 6/7 were obtained under dark conditions. This is now clarified (p14 lines 18–19): "Ozonolysis was performed under dark conditions without the addition of $NO_x$."

*7. Some typos should be corrected: page 5: line 3 "1/4" and 1/2"", line 17 and 18:"160W", "75W". Page 9: line 15: "4mm".*

We have now added spaces between the number and unit in each case (p5 line 29, p6 lines 14–15, p 11 line 25).

**References**

Anglada, J. M., Martins-Costa, M., Francisco, J. S. and Ruiz-lo, M. F.: Interconnection of Reactive Oxygen Species Chemistry across the Interfaces of Atmospheric, Environmental, and Biological Processes, Acc. Chem. Res., 48, 575–83, doi:10.1021/ar500412p, 2015.

Apel, K. and Hirt, H.: REACTIVE OXYGEN SPECIES : Metabolism, Oxidative Stress, and Signal Transduction, Annu. Rev. Plant Biol., 55, 373–99, doi:10.1146/annurev.arplant.55.031903.141701, 2004.

Badali, K. M., Zhou, S., Aljawhary, D., Antiñolo, M., Chen, W. J., Lok, a., Mungall, E., Wong, J. P. S., Zhao, R. and Abbatt, J. P. D.: Formation of hydroxyl radicals from photolysis of secondary organic aerosol material, Atmos. Chem. Phys., 15(14), 7831–7840, doi:10.5194/acp-15-7831-2015, 2015.

Camredon, M., Aumont, B., Lee-Taylor, J. and Madronich, S.: The SOA/VOC/NOx system : an explicit model of secondary organic aerosol formation, Atmos. Chem. Phys., 7, 5599–5610, 2007.

Chen, F., Zhou, H., Gao, J. and Hopke, P. K.: A Chamber Study of Secondary Organic Aerosol (SOA) Formed by Ozonolysis of d-Limonene in the Presence of NO, Aerosol Air Qual. Res., 17, 59–68, doi:10.4209/aaqr.2016.01.0029, 2017.

Chen, X. and Hopke, P. K.: A chamber study of secondary organic aerosol formation by limonene ozonolysis, Indoor Air, 20, 320–328, doi:10.1111/j.1600-0668.2010.00656.x, 2010.

Chen, X., Hopke, P. K. and Carter, W. P. L.: Secondary Organic Aerosol from Ozonolysis of Biogenic Volatile Organic Compounds: Chamber Studies of Particle and Reactive Oxygen Species Formation, Environ. Sci. Technol, 45(1), 276–282, 2011.

Docherty, K. S., Wu, W., Lim, Y. Bin and Ziemann, P. J.: Contributions of organic peroxides to secondary aerosol formed from reactions of monoterpenes with O3, Environ. Sci. Technol., 39(11), 4049–4059, doi:10.1021/es050228s, 2005.

Epstein, S. A., Blair, S. L. and Nizkorodov, S. A.: Direct Photolysis of  -Pinene Ozonolysis Secondary Organic Aerosol: Effect on Particle Mass and Peroxide Content, Environ. Sci. Technol, 48, 11251–58, 2014.

Gallimore, P. J., Achakulwisut, P., Pope, F. D., Davies, J. F., Spring, D. R. and Kalberer, M.: Importance of relative humidity in the oxidative ageing of organic aerosols: case study of the ozonolysis of maleic acid aerosol, Atmos. Chem. Phys., 11(23), 12181–12195, doi:10.5194/acp-11-12181-2011, 2011.

Tong, H., Arangio, A. M., Lakey, P. S. J., Berkemeier, T., Liu, F., Kampf, C. J., Brune, W. H., Poschl, U. and Shiraiwa, M.: Hydroxyl radicals from secondary organic aerosol decomposition in water, Atmos. Chem. Phys., 16(3), 1761–1771, doi:10.5194/acp-16-1761-2016, 2016.

Verma, V., Ning, Z., Cho, A. K., Schauer, J. J., Shafer, M. M. and Sioutas, C.: Redox activity of urban quasi-ultrafine particles from primary and secondary sources, Atmos. Environ., 43(40), 6360–6368, doi:10.1016/j.atmosenv.2009.09.019, 2009.

Wang, Y., Kim, H. and Paulson, S. E.: Hydrogen peroxide generation from a- and bpinene and toluene secondary organic aerosols, Atmos. Environ., 45(18), 3149–3156, doi:10.1016/j.atmosenv.2011.02.060, 2011.

Wragg, F. P. H., Fuller, S. J., Freshwater, R., Green, D. C., Kelly, F. J. and Kalberer, M.: An automated online instrument to quantify aerosol-bound reactive oxygen species (ROS) for ambient measurement and health-relevant aerosol studies, Atmos. Meas. Tech., 9, 4891–4900, doi:10.5194/amt-9-4891-2016, 2016.

---

## Author Response (AR1)

**Multiphase composition changes and reactive oxygen species formation during limonene oxidation in the new Cambridge Atmospheric Simulation Chamber (CASC)**

5 Peter J. Gallimore1, Brendan M. Mahon1, Francis P. H. Wragg1, Stephen J. Fuller1, Chiara Giorio1,2, Ivan Kourtchev1,3 and Markus Kalberer1

[revised manuscript text omitted]
 | Particle-phase chemical    | 0.2-600               |                       | 4-6 minutes   |
| Kalberer, 2013)        | composition                | $\mu g/m^3$           |                       |               |
| OPROSI (Wragg et al.,  | Particle-bound reactive    | 0-2000 nmol           | 2-4 nmol              | 4 minutes     |
| 2016)                  | oxygen species (ROS)       | $[H_2O_2]$            | $[H_2O_2]$            |               |
|                        |                            | equiv m -3 | equiv m -3 |               |
| Ionicon PTR-ToF 8000   | Gas-phase VOCs             |                       |                       | As low as 100 |
| MS                     |                            |                       |                       | ms, typically |
|                        |                            |                       |                       | 1 minute      |
| TSI 3086 SMPS          | Particle size distribution | 14-700 nm             |                       | < 2.5 minutes |
| Thermo 49C ozone       | [O 3 ]          | 0-200 ppm             | $\pm1$ ppb up to      | 1 minute      |
| analyser               |                            |                       | 1 ppm                 |               |
| Teledyne 200E NOx      | [NO], [NO2], [NOX]         | 0-1000 ppb            | $\pm 1 \text{ ppb}$   | 1 minute      |
| analyser               |                            |                       |                       |               |
| Sensirion SHT75        | RH, T                      | 0-100%                | ± 1.8%                | 1 second      |
|                        |                            | (RH), -40 –           | (RH), ±               |               |
|                        |                            | 120°C (T)             | 0.3°C (T)             |               |

Table 1: Overview of CASC instrumentation. EESI-MS and OPROSI are unique instruments developed in-house.

**10 2.2.1 Extractive Electrospray Ionisation Mass Spectrometry**

-Extractive Electrospray Ionisation Mass Spectrometery (EESI-MS) is an online particle analysis technique and the design and optimisation of our EESI source is described in Gallimore and Kalberer (2013). It contains a commercial electrospray probe (Thermo

8

- Formatted: Heading 3

Scientific HESI-II) with a custom-built aerosol injector and manifold. The primary electrospray was operated at a voltage of -3.0 kV and a N0 sheath flow rate of 0.8 L min-1. A water-methanol 1:1 mixture (Optima LC-MS grade solvents; Fisher Scientific) containing 0.05% formic acid (90%, Breckland Scientific) was infused into the ESI probe at 10  $\mu$ L min-1.

- 5 1. Chamber air was delivered into the source at 1 L min-1. Collision and extraction of the SOA particles by primary electrospray droplets occurs in an off-axis configuration with respect to the MS inlet, to minimise source contamination and memory effects through particle deposition. The EESI source was coupled to an ion-trap mass spectrometer (Thermo Scientific LTQ Velos). Spectra were acquired in the negative ionisation mode over the range
- 10  $\underline{m/z}$  50-500, with a mass resolution ~ 2000 (full width at half maximum, FWHM) at  $\underline{m/z}$  400.

Gallimore and Kalberer (Gallimore and Kalberer, 2013) demonstrated that the relative EESI-MS ion intensity correlated with the mass concentration of tartaric acid particles delivered into the source, suggesting that the entire particle bulk is extracted for analysis. More

- 15 recently, Gallimore et al., (Gallimore et al., 2017) showed that the kinetics of particle-phase reactions could be monitored; loss rates derived from EESI-MS measurements compared well with other studies, and spectra were compared to Liquid Chromatography (LC) MS to confirm that the EESI-MS assignments were present in the aerosol rather than formed as artefacts in the ion source.
- 20 It retains the key advantage of "soft" electrospray ionisation MS techniques, namely that quasi-molecular ions are produced from aerosol analytes with minimal fragmentation, and hence individual molecular species can be identified. It is also particularly suited to chamber measurements because time-resolved information is obtained and relative intensity changes can be linked to concentration changes in the particle Gallimore et al., (submitted).

**25**

**2.2.2 Proton Transfer Reaction Mass Spectrometry**

The gas phase VOC composition of the chamber is monitored using Proton Transfer Reaction MS (Blake et al., 2009). The PTR-MS (PTR-ToF 8000, Ionicon, Innsbruck, Austria) measures VOCs with a proton affinity higher than water in the *m/z* range 10-500, with a 30 typical mass resolution of 5000 (full width at half maximumFWHM) at the mass of

9

| - | Formatted: Subscript   |
|---|------------------------|
| + | Formatted: Superscript |
| + | Formatted: Superscript |
| - | Formatted: Superscript |

**Formatted: Font: Italic Formatted: Font: Italic**

protonated acetone, and a typical time resolution of 1\_s. Typical detection limits are in the order of 1-2\_ppb at 1\_s time resolution and ~30\_ppt at 1\_min time resolution (Blake et al., 2009; de Gouw and Warneke, 2007). For these experiments, source settings were: drift tube voltage of 600 V, drift tube pressure at ~ 2.20 mbar, drift tube temperature at 60°C, resulting
in an E/N of ca. 135 Td (1 Td = 10-17 V cm2). k = 2.54 × 10-9 cm3 molecule-1 s-1 was used for limonene quantification (Zhao and Zhang, 2004) and a default rate constant (k) of 2 × 10-9

**2.2.3 Online Particle-bound Reactive Oxygen Species Instrument**

cm3 molecule-1 s-1 was used for the other ions.

- 10 Reactive Oxygen Species (ROS) can beare associated with the negative health impacts of aerosols (den Hartigh et al., 2010; Steenhof et al., 2011). A new Online Particle-bound Reactive Oxygen Species Instrument (OPROSI), described in Wragg et al. (2016) is used to continuously monitor this health relevant property of aerosols from the chamber. The continuous sample inflow (5 L min-1) passes through a PM2.5 cyclone (URG-2000-30E-5-
- 15 2.5-S) and charcoal denuder prior to entering into a particle-into-liquid sampler (PILS). Particles are collected into a 1 mL min-1 spray containing horseradish peroxidase (HRP) (TypeVI, 1 unit mL-1 in 10% phosphate buffer solution (PBS), Sigma Aldrich) which reacts with ROS present in the particles. This is combined with a 1mL min-1 aqueous 2'7'dichlorofluorescein (DCFH) solution (10µM, 10% PBS, Sigma Aldrich), which is oxidised
- 20 to a fluorescent product (DCF) by the ROS-HRP solution. After a 10 minute reaction time at  $40^{\circ}$ C the concentration of DCF is quantified via fluorescence spectroscopy. The fluorescence response is calibrated with H2O2 and quantitative ROS concentrations are reported as "[H2O2] equivalents". The assay also responds to organic peroxides. It is likely sensitive to HO8 radicals and ions such as superoxide but we are unable to obtain suitable standards to test this
- 25 directly. OPROSI has a time resolution of 4 minutes (e-folding time during online particle collection tests) and is thus able to capture most time-dependant processes observed during SOA formation and evolution. This instrument is especially sensitive to short-lived ROS components, which react within seconds-minutes after sampling (Wragg et al., 2016)(see Wragg et al. (2016) for more details).

[revised manuscript text omitted]

condensed phase including peroxides, carbonyls and secondary ozonides (Lee et al., 2012; Maksymiuk et al., 2009).

**3. Limonene SOA formation and characterisation**

Insight into the chemical and health-relevant properties of limonene-derived SOA is provided
by the online characterisation techniques coupled to CASC. We focus on ozonolysis in order to compare the results from CASC with a range of previous studies which measure SOA chemical composition (Bateman et al., 2009; Kundu et al., 2012; Maksymiuk et al., 2009; Walser et al., 2008). In addition, from a human health perspective, exposure to limonene SOA is most likely to occur indoors, where ozone is the dominant sink of limonene.

10

Before the introduction of reactants, the concentrations of  $O_3$ ,  $NO_x$  and particles in the clean chamber were below the detection limits of the respective instruments in Table 1. The relative humidity of the chamber air was adjusted to 40 % and 6  $\mu$ L limonene (> 99%, Sigma) was added and mixed to produce a starting concentration of ~190 ppb based on PTR-MS

15 quantification. Ozone was introduced into the chamber over a 20 minute period, during which time the chamber air was regularly mixed; a maximum concentration of ~450 ppb was achieved (Figure 3). This corresponds to a stoichiometric excess of ozone with respect to the number of double bonds present in the limonene precursor. Ozonolysis was performed under dark conditions without the addition of NO&x

20

SOA was produced rapidly following the introduction of ozone to the chamber (Figure 3). Particles grew via homogeneous nucleation into a single mode with diameter ~160 nm and standard deviation  $\sigma$ =0.21. The measured SMPS data (black curve) were corrected for particle wall losses (red curve) using a procedure similar to Rollins et al., (2009) which is

25 described in the supplementary information. Over 85% of the loss-corrected mass was formed within the first 30 minutes of ozone introduction, with slower additional growth over the next  $\sim$  3 hours of the experiment.

---

## Author Response (AR2)

**Editor technical corrections:**

*Dear Authors:*

*Thank you for your detailed responses to the referee comments. I accept this manuscript for publication subject to the following technical corrections.*

*1) Page 20 line 22. I am assuming that Gallimore et al. submitted should instead by Gallimore et al. 2017.*

The reference now reads Gallimore et al., (2017).

*2) In Figures 1 and S3 please correct the structures to COO (limonaketone branch in Fig. 1).*

We have corrected these structures.